# Architecture of the centriole cartwheel-containing region revealed by cryo-electron tomography

Nikolai Klena[1,†] (ID), Maeva Le Guennec[1,†] (ID), Anne-Marie Tassin[2], Hugo van den Hoek[3,4],
Philipp S Erdmann[4], Miroslava Schaffer[4], Stefan Geimer[5], Gabriel Aeschlimann[6] (ID), Lubomir Kovacik[7],
Yashar Sadian[8], Kenneth N Goldie[7] (ID), Henning Stahlberg[7] (ID), Benjamin D Engel[3,4,9,*] (ID),
Virginie Hamel[1,**] (ID) & Paul Guichard[1,***] (ID)

## Abstract

Centrioles are evolutionarily conserved barrels of microtubule triplets that form the core of the centrosome and the base of the cilium. While the crucial role of the proximal region in centriole biogenesis has been well documented, its native architecture and evolutionary conservation remain relatively unexplored. Here, using cryo-electron tomography of centrioles from four evolutionarily distant species, we report on the architectural diversity of the centriole's proximal cartwheel-bearing region. Our work reveals that the cartwheel central hub is constructed from a stack of paired rings with cartwheel inner densities inside. In both *Paramecium* and *Chlamydomonas*, the repeating structural unit of the cartwheel has a periodicity of 25 nm and consists of three ring pairs, with 6 radial spokes emanating and merging into a single bundle that connects to the microtubule triplet via the D2-rod and the pinhead. Finally, we identified that the cartwheel is indirectly connected to the A-C linker through the triplet base structure extending from the pinhead. Together, our work provides unprecedented evolutionary insights into the architecture of the centriole proximal region, which underlies centriole biogenesis.

**Keywords** cartwheel; centriole; cryo-electron tomography; cryo-focused ion beam milling; *in situ*
**Subject Categories** Cell Adhesion, Polarity & Cytoskeleton; Structural Biology
**The EMBO Journal (2020) 39: e106246**

## Introduction

Centrioles and basal bodies (hereafter referred to as centrioles for simplicity) are cytoskeletal organelles, typically 450–550 nm in length and ~250 nm in outer diameter, which are present in most eukaryotic cells and play organizing roles in the assembly of cilia, flagella, and centrosomes (Nigg & Raff, 2009; Gönczy, 2012; Winey & O'Toole, 2014). Centrioles are characterized by a near-universal ninefold radial arrangement of microtubule triplets that contain a complete 13-protofilaments A-microtubule and incomplete B- and C-microtubules, each composed of 10 protofilaments (Guichard *et al*, 2013). Centrioles are polarized along their proximal-to-distal axis, with distinct structural features along their length. The proximal region is defined by the presence of the cartwheel structure, which serves as a seed for centriole formation and is thought to impart ninefold symmetry to the entire organelle (Nakazawa *et al*, 2007; Strnad & Gönczy, 2008; Gönczy, 2012; Hirono, 2014; Hilbert *et al*, 2016). In most species, the cartwheel stays within the centriole after maturation; however, it is not present in mature human centrioles (Azimzadeh & Bornens, 2007). The native architecture of the proximal region, and in particular of the cartwheel, was revealed by cryo-electron tomography (cryo-ET) of the *Trichonympha* centriole. Owing to its exceptionally long proximal region, many structural repeats could be sampled for subtomogram averaging, revealing the overall 3D structure of the cartwheel for the first time (Guichard *et al*, 2012, 2013). The *Trichonympha* cartwheel was observed to be built from a hub of stacked rings spaced every 8.5 nm. Radial spokes, emanating from two adjacent rings, merged at the pinhead near

1  Department of Cell Biology, University of Geneva, Sciences III, Geneva, Switzerland
2  Institute for Integrative Biology of the Cell (I2BC), CEA, CNRS, Univ. Paris Sud, Université Paris-Saclay, Gif sur Yvette, France
3  Helmholtz Pioneer Campus, Helmholtz Zentrum München, Neuherberg, Germany
4  Department of Molecular Structural Biology, Max Planck Institute of Biochemistry, Martinsried, Germany
5  Department of Cell Biology and Electron Microscopy, Universität Bayreuth, Bayreuth, Germany
6  Ribosome Studio Aeschlimann, Oberrieden, Switzerland
7  Center for Cellular Imaging and NanoAnalytics (C-CINA), Biozentrum, University of Basel, Basel, Switzerland
8  Bioimaging and Cryogenic Center, University of Geneva, Geneva, Switzerland
9  Department of Chemistry, Technical University of Munich, Garching, Germany
   *Corresponding author. Tel: +49 089 3187 1822; E-mail: ben.engel@helmholtz-muenchen.de
   **Corresponding author. Tel: +41 22 379 6735; E-mail: virginie.hamel@unige.ch
   ***Corresponding author. Tel: +41 22 379 6750; E-mail: paul.guichard@unige.ch
   †These authors contributed equally to this work.
   [Correction added on October 7 after first online publication: Projekt Deal funding statement has been added.]

the microtubule triplet to form a repeating structural unit with a periodicity of 17 nm. Moreover, this study demonstrated that each *Trichonympha* hub ring could accommodate nine homodimers of SAS-6, a protein that is essential for cartwheel assembly across eukaryotes (van Breugel *et al*, 2011, 2014; Kitagawa *et al*, 2011). Unexpectedly, a cartwheel inner density (CID), was also identified at the center of the hub ring. This CID contacts the hub ring at nine locations and has been hypothesized to be *Trichonympha*-specific, as CIDs have never been observed in other species, possibly due to lack of resolution. In this respect, the CIDs have been proposed to facilitate TaSAS-6 oligomerization or confer additional mechanical stability to these exceptional long centrioles, which are subjected to strong forces inside the intestine of the host termite (Guichard *et al*, 2013, 2018). Note that in Guichard *et al*, (2013), the abbreviation CID was defined as a connected circle of nine "cartwheel inner densities", but here we define this whole structure as a single CID to allow a clear description of our data.

In the proximal region, the cartwheel is connected to the pinhead, which bridges the cartwheel to the A-microtubule of the microtubule triplet (Dippell, 1968; Hirono, 2014). This connection is thought to be partially composed of Bld10p/Cep135 proteins, which can interact with both SAS-6 and tubulin (Hiraki *et al*, 2007; Carvalho-Santos *et al*, 2012; Kraatz *et al*, 2016; Guichard *et al*, 2017). In addition to the cartwheel/pinhead ensemble, adjacent microtubule triplets in the proximal region are also connected by the A-C linker. Cryo-ET combined with subtomogram averaging has revealed distinct structures of the A-C linker in *Trichonympha* and *Chlamydomonas reinhardtii* (Guichard *et al*, 2013; Li *et al*, 2019). In *Trichonympha*, the structure consists of the A-link, which is laterally inclined and contacts the A-tubule at the A8 protofilament, and the C-link, which connects to the C-tubule at the C9 protofilament. Overall, the *Trichonympha* A-C linker displays a longitudinal periodicity of 8.5 nm. In contrast, the A-C linker in *C. reinhardtii* is a crisscross-shaped structure composed of a central trunk region from which two arms and two legs extend to contact the A- and C-tubules (Li *et al*, 2019). Whereas these two studies provide major advances in our understanding of A-C linker organization, they also clearly highlight structural divergence between *Trichonympha* and *C. reinhardtii* centrioles.

The question thus arises as to the evolutionary conservation of the centriole's proximal region, including characteristic structures such as the A-C linker and the cartwheel's hub, CID, and radial spokes. In particular, the structure of the cartwheel remains unexplored beyond *Trichonympha*. A more universal description of the proximal region is important for understanding of how these structures direct centriole biogenesis. Here, we use cryo-ET to tackle this fundamental question using four evolutionarily distant species: *Chlamydomonas reinhardtii*, *Paramecium tetraurelia*, *Naegleria gruberi*, and humans.

## Results

### *In situ* structural features of the cartwheel in *Chlamydomonas* centrioles

The power of biodiversity proved extremely useful for resolving the first 3D architecture of the cartwheel within the exceptionally long proximal region of *Trichonympha* centrioles (Guichard *et al*, 2012). This study identified the CID as well as an 8.5 nm

longitudinal periodicity along the central hub of the cartwheel. Whether these structural features hold true in other species is an open question that we address here by analyzing the cartwheel of the green algae *C. reinhardtii*, a canonical model for centriole biology with similar centriole structure and protein composition to humans (Keller *et al*, 2005; Keller & Marshall, 2008; Li *et al*, 2011; Hamel *et al*, 2017). However, extracting centrioles from cells can limit the analysis of these fragile structures, as exemplified by the loss of the cartwheel during a study of isolated *C. reinhardtii* centrioles (Li *et al*, 2011). In addition, the > 300 nm thick vitreous ice surrounding uncompressed centrioles on an EM grid reduces the signal and contrast of cryo-ET (Kudryashev *et al*, 2012), making it difficult to resolve fine details in the relatively small cartwheel structure (Guichard *et al*, 2018). We therefore decided to analyze the *C. reinhardtii* cartwheel *in situ* using a cryo-focused ion beam (cryo-FIB) milling approach, which creates thin 100–150 nm sections of the native cellular environment in a vitreous state (Schaffer *et al*, 2017). Combining this approach with new direct electron detector cameras (Grigorieff, 2013), it was possible for us to visualize the centriole and cartwheel with unprecedented clarity and structural preservation.

As shown in Fig 1A and B, *in situ* cryo-ET clearly revealed both mature centrioles and procentrioles, providing the first observation of the centriole's cartwheel-bearing region in its native environment. The cartwheel's structural features were analyzed in both types of centrioles (Figs 1C–H and EV1 and Appendix Fig S1). Strikingly, we found that the cartwheel's central hub has an average longitudinal periodicity of 4.0 nm in both mature centrioles and procentrioles, distinct from the 8.5 nm periodicity originally described in *Trichonympha* (Guichard *et al*, 2012) (Figs 1H and EV1A, D and G). Moreover, we noticed pronounced densities inside the central hub that were reminiscent of the CIDs originally described in *Trichonympha*, suggesting that this structure is not *Trichonympha*-specific but rather is a conserved feature of the cartwheel (Fig 1). Several CIDs in *C. reinhardtii* are spaced along the lumen of the central hub, forming an 8.7 nm periodicity on average, in mature centrioles and procentrioles (Figs 1H–J and EV1B and E), similar to *Trichonympha*.

To investigate whether the discrepancy we observed in central hub periodicity was accompanied by other differences in cartwheel structure, we measured features of the cartwheel such as the central hub diameter and the distances from the hub to D1 and D2, two densities previously described on the cartwheel spokes of *C. reinhardtii* centrioles (Guichard *et al*, 2017) in both mature centrioles and procentrioles (Appendix Fig S2A–F). Similar to previous measurements, we found that the central hub is ~21 nm in diameter (peak-to-peak from the intensity plot profile through the hub), and the D1 and D2 densities are positioned ~36 nm and ~47 nm from the external edge of the cartwheel hub, respectively. These measurements suggest that only the longitudinal periodicity of the central hub differs in the *in situ C. reinhardtii* centrioles.

While most of the cartwheel's structural features, including the CIDs, are conserved between *Trichonympha* and *C. reinhardtii*, the periodicity of the central hub appears to diverge. This discrepancy poses the important question of how conserved the architecture of the cartwheel-containing region is between species. Moreover, as cartwheel periodicity was previously only measured in isolated centrioles, this raises the possibility that cartwheel periodicity may be affected during purification.

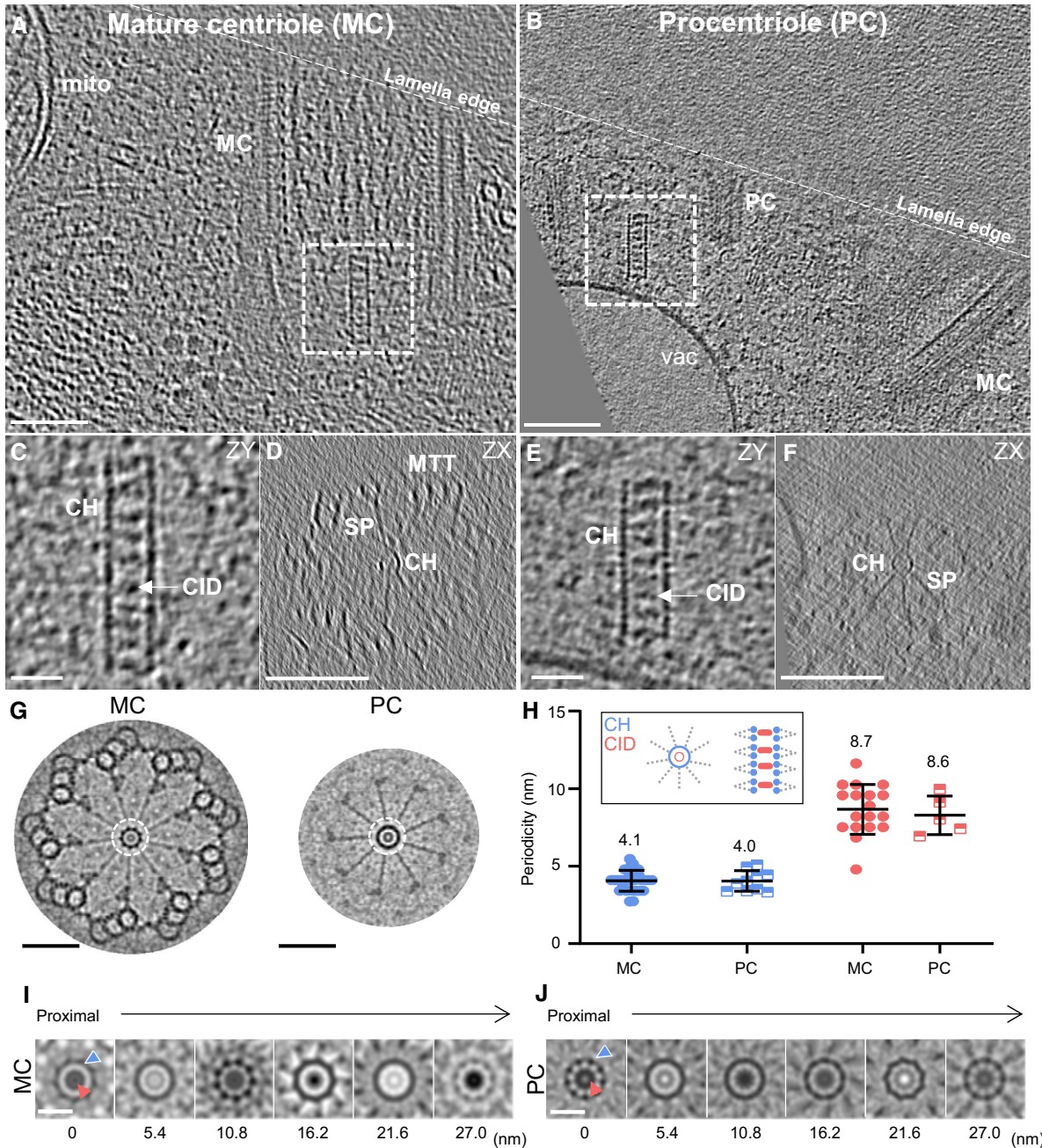

**Figure 1. *In situ* cryo-ET reveals the native cartwheel structure in *C. reinhardtii* centrioles.**

A, B    *In situ* cryo-electron tomogram displaying the proximal region of a mature mother centriole (A) and procentriole (B). Mature centriole, MC; procentriole, PC; mitochondria, mito; vacuole, vac; white dashed line, lamella edge. Scale bars, 100 nm.

C    Side view z-projection of cartwheels containing the central hub and several CIDs from a mature centriole. Central hub, CH; cartwheel inner density, CID. Scale bar, 20 nm.

D    Cross section of the cartwheel-containing region from a mature centriole. Microtubule triplet, MTT; spokes, SP. Scale bar, 200 nm.

E    Side view z-projection of a cartwheel containing the central hub and several CIDs from a procentriole. Scale bar, 20 nm.

F    Cross section of the cartwheel-containing region from a procentriole. Scale bar, 200 nm.

G    Ninefold symmetrized cross sections of the cartwheel-containing region from a mature centriole (left side) and a procentriole (right side). Dashed white circle, central hub. Scale bars, 100 nm.

H    Longitudinal periodicity measurements of the central hub and CIDs. Central hub, blue; CID, red. Mean values are displayed above the data range. Blue data points are measured distance between individual units of the central hub, and red data points are measured distances between individual units of the cartwheel inner density. Mature centriole, central hub, $n = 30$, mean = $4.1 \pm 0.67$ (SD); mature centriole, cartwheel inner density, $n = 18$, mean = $8.7 \pm 1.6$ (SD); procentriole, central hub, $n = 10$, mean = $4.0 \pm 0.66$ (SD); procentriole, cartwheel inner density, $n = 5$, mean = $8.6 \pm 1.2$ (SD).

I, J    Ninefold symmetrized central hub z-projections, starting at the proximal end of the cartwheel and continuing distally along the cartwheel by 5.4 nm steps in a mature centriole (I) and a procentriole (J). Red arrow, CID; blue arrow; central hub. Scale bar: 20 nm.

## Conservation of the cartwheel's structural features in *Paramecium*, *Naegleria*, and humans

To address these questions, we analyzed the proximal region of isolated centrioles from three different species. Centrioles were purified from *P. tetraurelia*, *N. gruberi* and human KE37 leukemia acute lymphoblastic T cells, vitreously frozen onto EM grids, and then imaged by cryo-ET (Figs 2A–I and EV2). Despite the high level of noise expected in cryo-ET of isolated centrioles, as well as the previously observed strong compression of *N. gruberi* and human centrioles (Guichard *et al*, 2010; Greenan *et al*, 2018; Le Guennec *et al*, 2020) that affects cartwheel integrity, we could reliably measure the central hub periodicity in each of these species. Strikingly, we found that the longitudinal periodicity of the central hub is similar to the *C. reinhardtii in situ* cartwheel, with average periodicities of 4.3 ± 0.38 nm, 4.4 ± 0.53 nm, and 4.2 ± 0.68 nm in *P. tetraurelia*, *N. gruberi*, and human, respectively (Figs 2J and EV2). Moreover, we observed that CID structures are present in every species, forming a periodicity along the central hub of 8.4 ± 1.25 nm, 8.3 ± 1.83 nm, and 8.1 ± 2.46 nm (Fig 2A–J and Appendix Fig S2G–O). These results indicate that structural features of the *C. reinhardtii* cartwheel seem to be conserved, including the central hub's ~4.2 nm periodicity, as well as the presence of CIDs every ~8.4 nm. Moreover, these measurements demonstrate that the discrepancy between *Trichonympha* and *C. reinhardtii* is probably not due to purification artifacts, as the other isolated centrioles also display ~4 nm periodicities along their central hubs.

Interestingly, in tomograms of both *in situ* and isolated centrioles, we observed that the position of the cartwheel did not fully correlate with the position of the microtubule triplets. In all four species, the cartwheels protruded proximally 10–40 nm beyond the microtubule wall (Figs 1A and B, and 2K and L). In *C. reinhardtii*, which enabled observations of assembling and mature centrioles within the same cells, the cartwheel extension was more prominent in procentrioles, with 67% of the cartwheel protruding in contrast to 34% in mature centrioles (Fig 2K). Until now, this proximal extension of the cartwheel has only been reported in isolated *C. reinhardtii* procentrioles (Geimer & Melkonian, 2004; Guichard *et al*, 2017). Our *in situ* C. *reinhardtii* tomograms demonstrate that the cartwheel extension is not an artifact of purifying centrioles, but rather occurs within the native cellular environment. We further corroborated this conclusion with serial sections of resin-embedded *N. gruberi* cells, which show the cartwheel protruding beyond the proximal end of the microtubule triplets in both assembling and mature centrioles (Appendix Fig S3). Interestingly, by applying a ninefold circularization on the cryo-tomograms of *P. tetraurelia* and *C. reinhardtii* (Fig EV3), we observed that the spokes emanating from the cartwheel proximal extension are organized similarly to the cartwheel region surrounded by microtubules. Moreover, we could identify that the extremities of the spokes are connected together vertically via the D2 densities (which we name the D2-rod) without any pinhead density visible, both in procentrioles and mature centrioles (Fig EV3).

The cartwheel proximal extension is consistent with fluorescence microscopy localization of cartwheel components CrSAS-6 and Bld10p, which extend from the centriole's proximal region to < 60 nm below the proximal-most acetylated tubulin signal in mature *C. reinhardtii* centrioles (Hamel *et al*, 2017). Additionally, this proximal extension corroborates 3D-SIM-FRAP analysis of SAS-6-GFP in *Drosophila*, showing that the cartwheel may grow from its proximal end (Aydogan *et al*, 2018). Taking these data together, we conclude that the cartwheel protrusion is not a consequence of biochemical isolation but rather is an evolutionarily conserved structural feature that may relate to early events in centriole assembly.

### 3D architecture of the cartwheel in *Paramecium* and *Chlamydomonas*

Given the intriguing ~4 nm periodicity of the central hub revealed in our study, which differs from the previously reported periodicity in *Trichonympha* (Guichard *et al*, 2012), we decided to take a closer

**Figure 2. Cryo-ET of isolated centrioles from *P. tetraurelia*, *N. gruberi*, and *H. sapiens* reveals novel cartwheel periodicities.**

A–C   Cryo-electron tomograms of the proximal regions of a *P. tetraurelia* centriole (A), a *N. gruberi* centriole (B), and a *H. sapiens* procentriole (C). White arrows denote a broken cartwheel; procentriole, PC; mature centriole, MC; Scale bar, 100 nm. Note that most *N. gruberi* and *H. sapiens* centrioles were heavily compressed during the cryo-EM preparation, as previously described (Guichard *et al*, 2010). The displayed *N. gruberi* centriole illustrates the damage caused by compression. The periodicities of *N. gruberi* and *H. sapiens* cartwheels were measured only on regions that were not damaged (see Fig EV2).

D   Cross section from cartwheel-containing region of a *P. tetraurelia* centriole. Scale bar, 50 nm.

E   Zoomed side view of cartwheel from *P. tetraurelia*, displaying the central hub (CH) and several cartwheel inner densities (CIDs), white arrow. Scale bar, 25 nm.

F   Cross section from cartwheel-containing region of a *N. gruberi* centriole. Same scale bar as in (D).

G   Zoomed side view of cartwheel from *N. gruberi*, displaying the central hub (CH) and several cartwheel inner densities (CIDs), white arrow. Same scale bar as in (E).

H   Cross section from cartwheel-containing region of a *H. sapiens* centriole. Same scale bar as in (D).

I   Zoomed side view of cartwheel from *H. sapiens*, displaying the central hub (CH) and several cartwheel inner densities (CIDs), white arrow. Same scale bar as in (E).

J   Longitudinal periodicity of the central hub and CIDs in *P. tetraurelia*, *N. gruberi*, and *H. sapiens*. Mean values are displayed above data range. Black lines indicate the mean and the standard deviation. Blue data points are measured distance between individual units of the central hub, and red data points are measured distances between adjacent cartwheel inner densities. *P. tetraurelia*, CH, n = 10, mean = 4.3 ± 0.38 (SD); *N. gruberi*, CH, n = 10, mean = 4.4 ± 0.53 (S.D); *H. sapiens*, CH, n = 10, mean = 4.2 ± 0.68 (S.D); *P. tetraurelia*, CID, n = 8, mean = 8.4 ± 1.3 (SD); *N. gruberi*, CID, n = 8, mean = 8.3 ± 1.8 (SD); *H. sapiens*, CID, n = 8, mean = 8.1 ± 2.5 (SD).

K, L   Proximal protrusion length of the cartwheel beyond the microtubule triplets in *C. reinhardtii*, *P. tetraurelia*, *N. gruberi*, and *H. sapiens*. Internal cartwheel inside the microtubule barrel, dark blue (INT); external cartwheel beyond the microtubule wall, light blue (EXT). Mean values are displayed above each bar plot, with the black lines indicating the standard deviation (K). The start of each microtubule wall is delineated by a dashed white line (L). Mature *C. reinhardtii*, n = 4, external cartwheel length = 37.6 ± 3.4, internal cartwheel length = 74.2 ± 16.6; *C. reinhardtii* procentriole, n = 2, external cartwheel length = 52.8 ± 22.0, internal cartwheel length = 26.5 ± 29.7; *P. tetraurelia*, n = 23, external cartwheel length = 11.5 ± 9.2, internal cartwheel length = 66.6 ± 15.7; *N. gruberi*, n = 19, external cartwheel length = 41.0 ± 22.5, internal cartwheel length = 259.0 ± 87.2; human procentriole, n = 7, external cartwheel length = 23.1 ± 10.3, internal cartwheel length = 154.7 ± 47.7. Reported values are mean and errors are standard deviation. Scale bar, 50 nm.

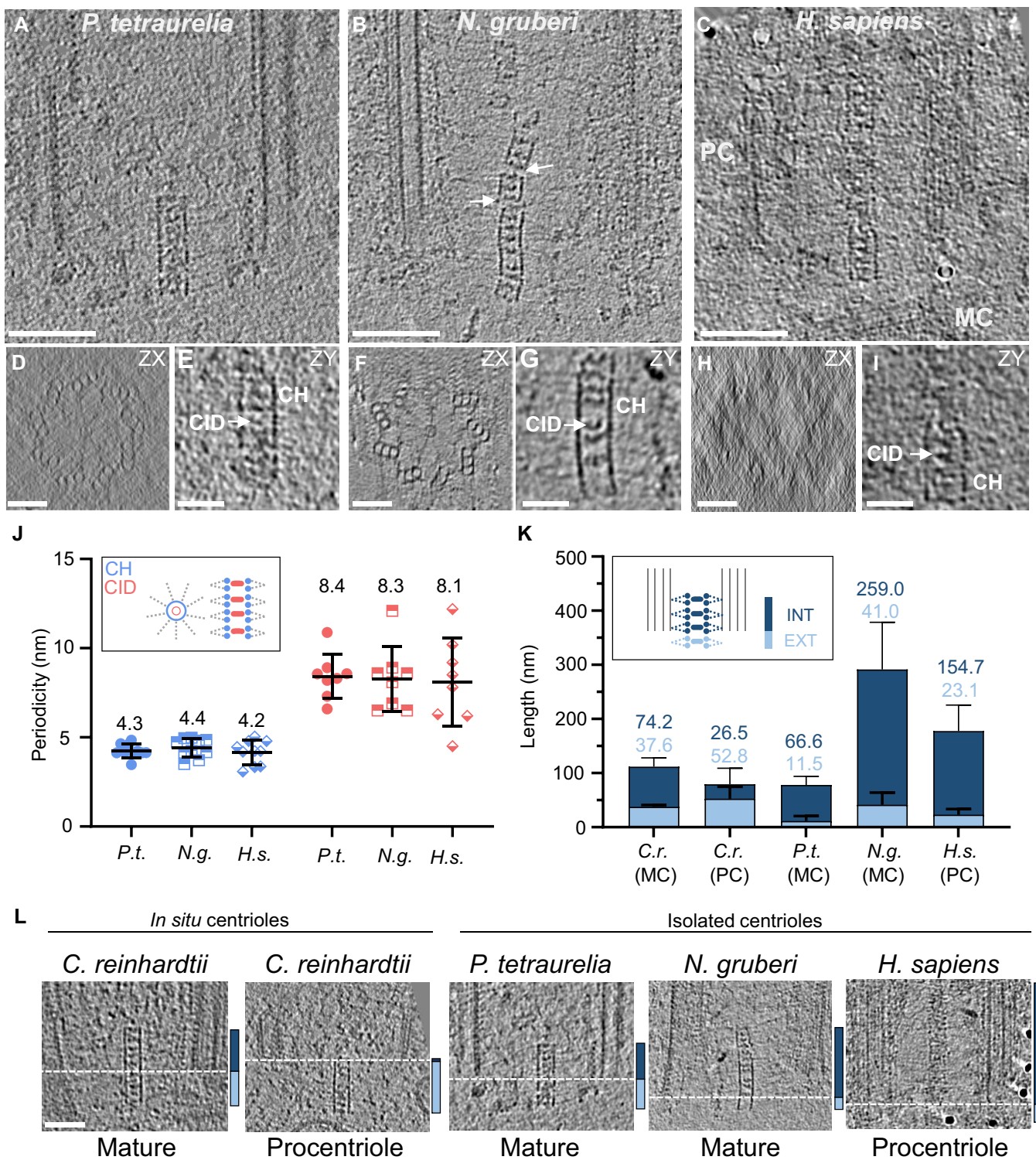

**Figure 2.**

look at the cartwheel architecture in both *P. tetraurelia* and *C. reinhardtii* centrioles. As explained above, resolving the cartwheel structure in these species represents a major challenge, as the cartwheel length is about 40 times shorter than the exceptionally long *Trichonympha* cartwheel, limiting the number of repeat units

available for subtomogram averaging. Nevertheless, we undertook this task with a low number of subvolumes, increasing the contrast of the central hub and emanating radial spokes. From 8 *P. tetraurelia* tomograms, we performed subtomogram averaging on 235 boxes and symmetrized the obtained map. A projection of the

reconstructed *P. tetraurelia* cartwheel is shown in Fig 3A, where the CID, the central hub, its emanating radial spokes, and the D2-rod are clearly visible. Careful inspection of a longitudinal section through the averaged volume confirmed the presence of CIDs every 8.6 nm inside the central hub (Figs 2J and 3C). Intriguingly, we found that the central hub is constructed from pairs of rings (Fig 3B and C, light blue arrowheads). These ring pairs have an inter-ring distance of 3.1 nm and stack on each other with 5.5 nm between adjacent ring pairs, resulting in the average periodicity of ~4.2 nm along the central hub (Figs 2J and 3C). We observed that two small densities (Fig 3D, white arrows) emanate from each ring pair (blue circles) and fuse into one radial spoke (white arrowheads), which in turn merges with two other fused spokes to form a single structure ~37 nm from the central hub surface, a distance that corresponds to the D1 density (Fig 3D, black arrow; Fig EV4A, black arrows). The D1 density connects to the D2-rod density (dark orange arrow) ~46 nm from the central hub surface. The three ring pairs that share fused spokes are repeated three to four times along the cartwheel length, with a longitudinal distance of ~25 nm between D1 merged spoke densities (Figs 3B and EV4A, black arrows), suggesting that this represents the repeating structural unit of the cartwheel. We also noted that the emanating spokes are slightly tilted in *P. tetraurelia* (white dashed lines in Figs 3D, J and EV4A), possibly reflecting a twist in the molecular interaction underlying spoke fusion. Interestingly, we found that the CIDs are positioned at the center of each ring pair (Fig 3C), suggesting that they could be important for the ring pair's formation or stability. Importantly, all these features can also be seen within the raw data (Appendix Fig S4A and B), indicating that they are not a result of the averaging procedure.

Next, we performed a similar analysis on *C. reinhardtii* mature centrioles (Fig 3E), using 102 subvolumes from 5 *in situ* tomograms and then applied symmetrization. Interestingly, we found that the cartwheel's repeating structural unit is also composed of three ring pairs, with 3.5 nm inter-ring spacing and 5.1 nm spacing between ring pairs (Fig 3E–G, blue arrowhead in G), leading to the observed ~4.2 nm periodicity along the central hub. Each repeating unit also had six emanating spokes (Figs 3H and EV4B, white arrows); however, these spokes were organized differently than in *P. tetraurelia* cartwheels, merging into two spokes ~17 nm from the central hub (Figs 3H and EV4B, white arrowheads), further fusing into a single D1 unit ~37 nm from the hub (Figs 3H and EV4B, black arrows), and extending to the D2-rod density (dark orange arrow) ~46 nm from the hub. Similar to *P. tetraurelia*, the repeating unit of the central hub has a periodicity of ~25 nm (Figs 3F and EV4B, black arrows). In *C. reinhardtii* cartwheels, CIDs are positioned 8.8 nm apart, inside ring pairs (Fig 3G). As for *P. tetraurelia*, we confirmed that these *C. reinhardtii* features could be seen in the raw data (Appendix Fig S4C and D) and were not a result of the averaging. We also noticed in raw tomograms that some regions were devoid of CIDs, suggesting that their positioning might be stochastic (Appendix Fig S4D, white arrowhead).

Together, these results demonstrate that both species have an overall similar cartwheel organization, with some species-specific differences in the radial spokes that possibly reflect either a different modality of assembly or some divergence at the molecular level. Moreover, we noticed that the repeating structural unit described here displays a polarity from proximal to distal that is defined by the angle of the emanating spokes, which is strikingly apparent in the *P. tetraurelia* average (Fig 3I and J, Movie EV1) and also

**Figure 3. Subtomogram averaging of *Paramecium* and *Chlamydomonas* cartwheels reveals novel cartwheel structural organization.**

A  Top view of cartwheel reconstruction from *P. tetraurelia*. Scale bar, 50 nm. Dark orange arrow marks the D2-rod. Purple arrow marks part of the pinhead. Dotted yellow box with arrows denotes the central hub-focused reslice shown in panel B; dashed white box with arrows denotes the spoke-focused reslice shown in panel D.

B  Reslice of central hub-containing region with spokes in *P. tetraurelia*. Scale bar, 50 nm. Dashed light yellow line denotes the zoomed view shown in panel C, black line with arrows indicates the ~25 nm repeat distance between D1 densities formed by merged spokes. Dark orange arrow marks the D2-rod.

C  Zoomed view displaying periodic repeats of the central hub (CH) and several cartwheel inner densities (CIDs) in *P. tetraurelia*. CIDs, red arrowheads and red plot profile; CH, blue arrowheads and blue plot profile. Overlay between CID and CH peaks is plotted on the right in red and blue. Mean distance between CID peaks, red 8.6 ± .8 (*n* = 5). Distances between CH peaks split into two distinct populations: smaller (within a ring pair), green 3.1 ± 1.3 (*n* = 6); larger (between ring pairs), blue 5.5 ± .7 (*n* = 5); average periodicity, black 4.2 ± 1.3 (*n* = 11).

D  Serial z-projections of ~4 nm thickness from one cartwheel repeat unit of *P. tetraurelia*. Left-most z-projections display the central hub, and right-most projection shows the microtubule wall. Blue circles delineate one ring pair. White arrows mark individual emanating spokes. White arrowheads mark fused spokes. White dashed line follows the tilt of the spokes. Black arrow indicates the final merged spoke (D1 density). Dark orange arrow marks the D2-rod. Scale bar, 50 nm.

E  Top view of cartwheel reconstruction from *C. reinhardtii*. Scale bar, 50 nm. Dark orange arrow marks the D2-rod. Purple arrow marks part of the pinhead. Dotted yellow box with arrows denotes the central hub-focused reslice shown in panel F; dashed white box with arrows denotes the spoke-focused reslice shown in panel H.

F  Reslice of central hub-containing region with spokes in *C. reinhardtii*. Scale bar, 50 nm. Dashed light yellow line denotes the zoomed view shown in panel G, black line with arrows indicates the ~25 nm repeat distance between D1 densities formed by merged spokes. Dark orange arrow marks the D2-rod.

G  Zoomed view displaying periodic repeats of the central hub (CH) and several cartwheel inner densities (CIDs) in *C. reinhardtii*. CIDs, red arrowheads and red plot profile; CH, blue arrowheads and blue plot profile. Overlay between CID and CH peaks is plotted on the right in red and blue. Mean distance between CID peaks, red 8.8 ± 1.0 (*n* = 5). Distances between CH peaks split into two distinct populations: smaller (within a ring pair), green 3.5 ± 1.3 (*n* = 6); larger (between ring pairs), blue 5.1 ± 0.4 (*n* = 5); average periodicity, black 4.2 ± 1.3 (*n* = 11).

H  Serial z-projections of ~4 nm thickness from one cartwheel repeat unit of *C. reinhardtii*. Left-most z-projections display the central hub, and right-most projection shows the microtubule wall. Blue circles delineate one ring pair. White arrows mark individual emanating spokes. White arrowheads mark fused spokes. White dashed line follows the tilt of the spokes. Black arrow indicates the final merged spoke (D1 density). Dark orange arrow marks the D2-rod. Scale bar, 50 nm.

I  Three-dimensional rendering of the cartwheel reconstruction from *P. tetraurelia*. Left side, cartwheel oriented along the correct proximal–distal axis; right side, inverted proximal–distal axis, showing the asymmetry of spoke inclination. Dark orange arrows indicate the D2-rod. Purple arrows indicate part of the pinhead. Dashed yellow box, inset of one spoke unit, with the major and minor tilt angles of the spokes relative to the central hub. White asterisks denote subunits of ring pairs. For three-dimensional rendering of the *C. reinhardtii* cartwheel, see Fig EV4C.

J  Model of *P. tetraurelia* (left side) and *C. reinhardtii* (right side) cartwheel structures. Dashed gray box denotes one repeat unit of the cartwheel, dashed black lines, and boxes display cross sections of spokes.

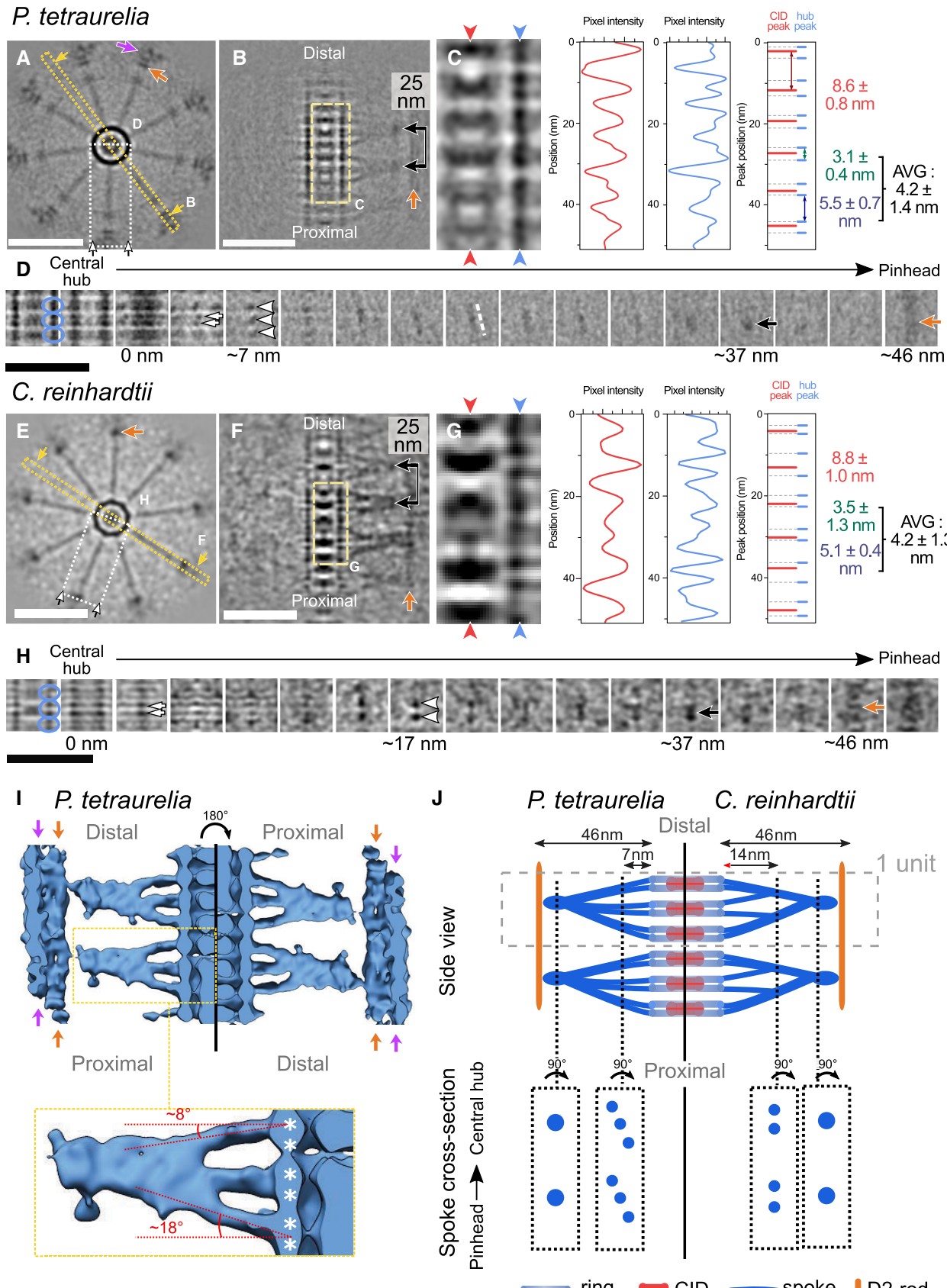

**Figure 3.**

distinguishable in the *C. reinhardtii* average despite its lower resolution (Fig EV4C).

Next, we investigated how the observed discrepancy in central hub periodicity could arise between *C. reinhardtii*/*P. tetraurelia* and *Trichonympha*. We hypothesized that the resolution improvement from using a direct electron detector might have helped reveal features that were not visible in the previous study of *Trichonympha* centrioles. To test this idea, we applied a bandpass filter to decrease the resolution of the *P. tetraurelia* subtomogram average to that of the *Trichonympha* map (38 Å) (Fig EV4D and E). At this resolution, the *P. tetraurelia* ring pairs appear to be single rings, leading to a global 8.6 nm periodicity along the central hub as originally described in *Trichonympha*. This result corroborates the observation made by the accompanying Nazarov *et al* manuscript (Nazarov *et al* 2020) that the *Trichonympha* cartwheel exhibits the same ~4 nm ring pair periodicity as *P. tetraurelia* and *C. reinhardtii*. This conserved ~4 nm periodicity could not be retrieved in earlier studies primarily due to resolution limitations of the detectors used for imaging. However, we also noticed that the spoke organization appears different between *Trichonympha* and *C. reinhardtii*/*P. tetraurelia* cartwheels, suggesting variability of molecular organization between species.

**Defining the structural features of the proximal region**

We next focused on charting the overall organization of the cartwheel-containing region in *P. tetraurelia* and *C. reinhardtii* centrioles to better understand how the cartwheel is connected to the microtubules and to check whether the structural features are conserved between species (Fig 4). As subtomogram averaging might average out non-periodic structures, we first analyzed the raw tomograms by systematically extracting cross sections of centrioles from both species at different positions along the proximal-to-distal axis and then applying ninefold symmetrization to improve the contrast using centrioleJ (Guichard *et al*, 2013) (Fig 4A, B, G and H from panel B). Starting from the proximal side, several previously described structural features could be resolved, including the cartwheel (blue arrow), the pinhead (purple arrow), the A-C linker (turquoise arrow), and the beginning of the inner scaffold (orange arrow) that defines the central core region of the centriole (Fig 4B, C, F, H and I). We also noticed a linker between the pinhead structure and the A–C linker (Fig 4C panels (III, IV) and 4I panels (III, IV), light green arrow). This linker is reminiscent of the triplet base structure originally described in human, mouse, and Chinese hamster centrioles (Vorobjev & Chentsov, 1980) and also detected in *Trichonympha* centrioles (Gibbons & Grimstone, 1960). We therefore conclude that the triplet base is an evolutionarily conserved structural feature of the centriole's cartwheel-bearing region. Interestingly, in contrast to the A-C linker (Fig 4C, panel (VI) and 4I, panel (VI)), the pinhead structure does not co-exist with the inner scaffold, suggesting that the latter replaces the former (Fig 4D, E, J–N and Appendix Fig S5A and D). In the most distal part of the proximal region, we also noticed that the pinhead structure is present without the cartwheel in *P. tetraurelia* centrioles (Fig 4B and C panel (IV), D and E, and Appendix Fig S5A and D). Finally, we observed in the two *in situ C. reinhardtii* procentrioles that the A-C linker covers the entire length of the growing microtubule triplets, while the pinhead and cartwheel display variable lengths (Appendix Fig S5G).

On the basis of these observations, we measured the distance from the end of the pinhead region to the end of the cartwheel region and to the start of the inner scaffold in 5 *in situ C. reinhardtii* centrioles and 17 isolated *P. tetraurelia* centrioles. We found that the distances between these structural features are ~5 nm on average in *C. reinhardtii*, which is close to the size of a tubulin monomer, indicating a direct transition from one structure to the other (Appendix Fig S5E and F). In contrast, this gap distance is longer and more variable in *P. tetraurelia* centrioles, suggesting more stochasticity in the transitions between structures (Appendix Fig S5E and F). We also noted a strong correlation between the lengths of the A–C linker and the pinhead in *P. tetraurelia* centrioles (Appendix Fig S5B), suggesting that these two structures might have coordinated assembly. Conversely, there is no clear correlation between the

---

**Figure 4. Structural features of the centriole's proximal region in *P. tetraurelia* and *C. reinhardtii*.**

A Cryo-electron tomogram of *P. tetraurelia* centriole. Blue arrow denotes cartwheel, and orange arrow denotes inner scaffold. White dashed box delimits the inset represented in (D, E). Red lines and arrows indicate the position and the direction of the cross sections made in (B). Scale bar, 100 nm.

B Ninefold symmetrizations of serial cross sections taken along the proximal to distal axis in *P. tetraurelia*. Each section is a z-projection of 20.7 nm. White dashed circles delineate the structures highlighted in C. Scale bar, 60 nm.

C Zoomed images from panel B of proximal centriole substructures from ninefold symmetrizations of *P. tetraurelia* along the proximal–distal axis. Each panel corresponds to the above image from panel B. Purple arrow, pinhead; light green arrow, triplet base; turquoise arrow, A-C linker; orange arrow, inner scaffold. Scale bar, 50 nm.

D Side view showing the transition from pinhead to inner scaffold in *P. tetraurelia*. Scale bar, 100 nm.

E Cartoon representation of panel D.

F Representative model of a cross section of a centriole's proximal region. Colored arrows indicate the different structural features identified.

G Cryo-electron tomogram of *C. reinhardtii* centriole. Blue arrow denotes cartwheel, and orange arrow denotes inner scaffold. White dashed box delimits the inset represented in (J, K). Red lines and arrows indicate the position and the direction of the cross sections made in (H). Same scale bar as in (A).

H Ninefold symmetrizations of serial cross sections taken along the proximal to distal axis in *C. reinhardtii*. Each section is a z-projection of 20.7 nm. White dashed circles delineate the structures highlighted in I. Scale bar, 60 nm.

I Zoomed images of proximal centriole substructures from ninefold symmetrizations of *C. reinhardtii* along the proximal–distal axis. Each panel corresponds to the above image from panel H. Purple arrow, pinhead; light green arrow, triplet base; turquoise arrow, A-C linker; orange arrow, inner scaffold. Scale bar, 50 nm.

J, K Side views showing the transition from pinhead to inner scaffold in *C. reinhardtii* from two consecutive sections from a tomogram. Scale bar, 100 nm.

L Cartoon representation combining the z-projections in panels J and K.

M, N Positioning of the different structures along the proximal length from representative *P. tetraurelia* (M) and *C. reinhardtii* (N) centrioles. Distance between the ends of the pinhead and cartwheel regions is denoted by zone 1 (for quantification, see Fig EV4E). Distance between end of the pinhead region and start of the inner scaffold region is denoted by zone 2 (for quantification, see Fig EV4F).

O Cartwheel and A-C linker length in *C. reinhardtii* (*n* = 5 centrioles), *P. tetraurelia* (*n* = 16 centrioles), *N. gruberi* (*n* = 11 centrioles), and *H. sapiens* (*n* = 5 centrioles). Means and standard deviations of the mean are displayed above the range. A-C linker, turquoise; cartwheel, blue; microtubule triplets, gray.

---

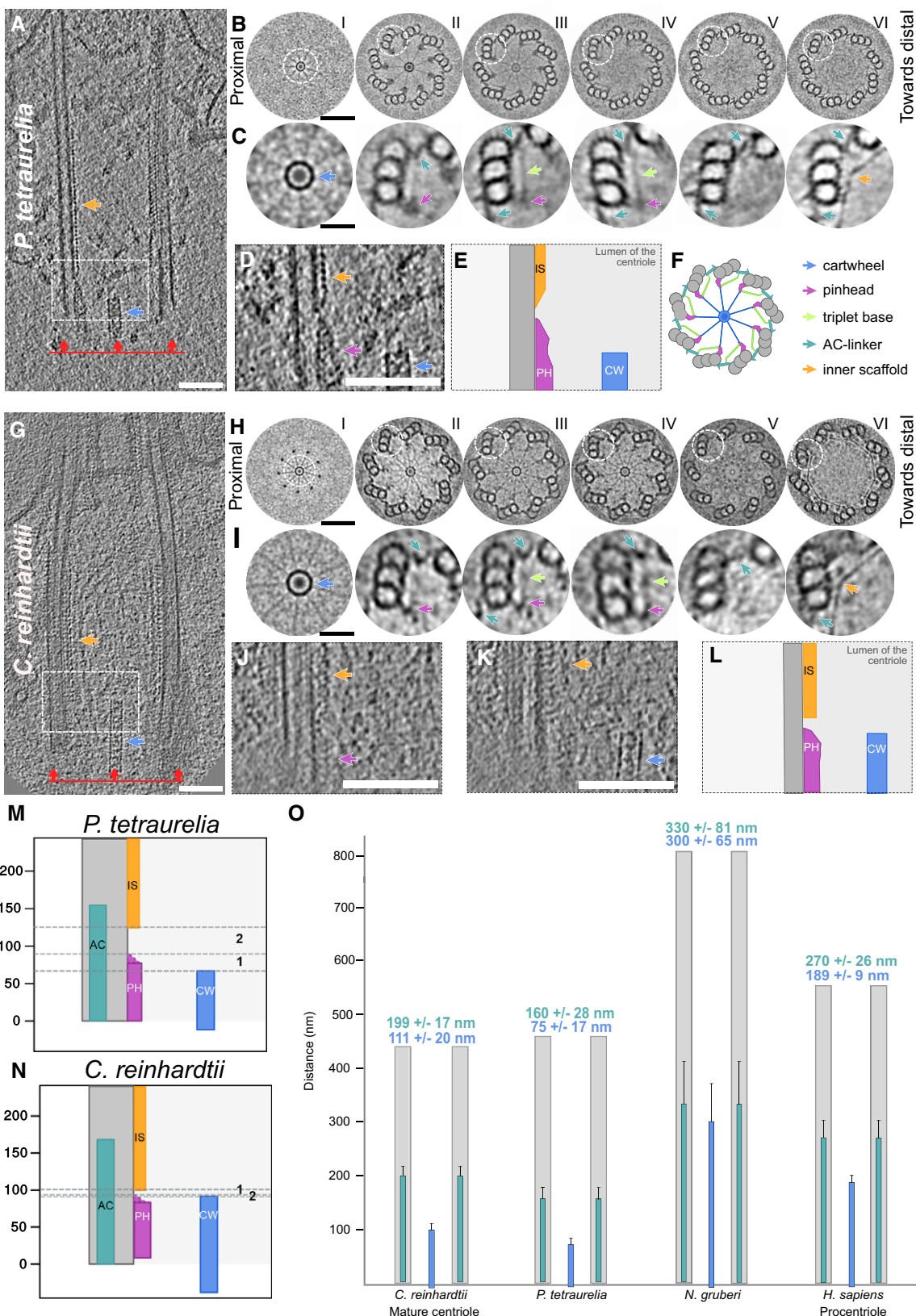

lengths of the cartwheel and pinhead in *P. tetraurelia* centrioles (Appendix Fig S5C).

To better understand the relationship between the A–C linker and the cartwheel, we mapped their respective boundaries in the centrioles of *P. tetraurelia*, *C. reinhardtii*, *N. gruberi,* and humans (Fig 4O). We found that the cartwheel length extends $111 \pm 20$ nm, $75 \pm 17$ nm, $300 \pm 65$ nm, and $189 \pm 9$ nm in *C. reinhardtii*, *P. tetraurelia*, *N. gruberi,* and humans, respectively (Fig 4O). Note that, as expected, mature human centrioles lacked cartwheels (Guichard *et al*, 2010), but we found 4 procentriole cartwheels to include in our analysis. In parallel, we analyzed the boundaries of the A-C linker and found that it spans $199 \pm 17$ nm, $160 \pm 28$ nm, $330 \pm 81$ nm, and $270 \pm 26$ nm of the proximal region in *C. reinhardtii*, *P. tetraurelia*, *N. gruberi,* and humans, respectively (Fig 4O). As previously reported (Le Guennec *et al*, 2020), this represents approximately 40% of the total centriole length. Comparing the measurements of these two structures reveals that the cartwheel spans 56% of the A-C linker length in *C. reinhardtii*, 47% in *P. tetraurelia*, 66% in *N. gruberi*, and 70% in humans.

### The triplet base bridges the pinhead with the A-C linker

Our analysis of raw tomograms revealed that the triplet base emanates from the pinhead and binds the A-C linker, thereby indirectly connecting the cartwheel to the A-C linker (Fig 4). However, this analysis did not allow us to precisely detect where the triplet base connects to the A-C linker. Moreover, this connection has never been observed in previous subtomogram averaging analysis (Guichard *et al*, 2013; Li *et al*, 2019). Consequently, we undertook a subtomogram averaging approach focused on revealing the triplet base connection and the A-C linker structure, using 11 tomograms of uncompressed *P. tetraurelia* centrioles. We succeeded in resolving the triplet base in our average; however, it had very low map density, suggesting that this structure is flexible or not stoichiometrically occupied (Fig 5A) and explaining why it has not been observed before in cryo-ET. It is also important to note that although both the triplet base and the pinhead are clearly visible, we could not reliably retrieve their longitudinal periodicities.

Next, we focused on the A-C linker and found that it can be subdivided into two major regions previously observed in *Trichonympha*: the A-link that contacts the A-tubule and the C-link that contacts the C-tubule. The *P. tetraurelia* A-C linker has a longitudinal periodicity of $8.4 \pm 0.2$ nm, consistent with previous measurements from *Trichonympha* and *C. reinhardtii* (Guichard *et al*, 2013; Li *et al*, 2019) (Fig EV5). With the obtained resolution of 31.5 Å (Appendix Fig S6), we were able to identify that the C-link is composed of two main densities: ArmA, which contacts the C-tubule protofilaments C8 and C9, and ArmB, which decorates only C-tubule protofilament C9 (Figs 5B–D and EV5). On the A-link side, we identified a single connection between the A-link's trunk and A-tubule protofilament A8, an interaction originally described in *C. reinhardtii* (Li *et al*, 2019) (Fig EV5F and G). In addition to the A-C linker, we identified a large density between protofilaments A8 and A9 of the A-tubule that we termed the A-tusk (Figs 5B–D and EV5C–E). Interestingly, we observed that the triplet base connects to the A-C linker directly on the ArmB density (Figs 5A and EV5D), reinforcing our conclusion that the entire proximal region forms an interconnected structural network

from the central hub of the cartwheel, through the radial spokes, the pinhead, and the triplet base to the A-C linker.

To check whether the connection between the pinhead and the A-C linker is maintained throughout the proximal region, we split the dataset in two halves corresponding to the more proximal and more distal parts of this region (Fig 5E and F and Appendix Fig S7). The ninefold symmetrized model of each map was reconstructed. Interestingly, as previously observed (Fig 4B and C), we noticed that the pinhead density is almost completely absent in the average from the more distal part of the proximal region, whereas the A-C linker is still present and has an extra density on ArmB seemingly replacing the triplet base position (Fig 5E and F, red circles). This observation indicates that although the pinhead and A-C linker are connected through the triplet base, the presence of the A-C linker is independent of the pinhead and triplet base. We also noticed a difference in the microtubule triplet and A-C linker angles between the two maps (Fig 5F), with an angle decrease of 6° for the triplet and 9° for the A-C linker. As this difference was previously observed in *C. reinhardtii* (Li *et al*, 2019), the slight twist we measured in the proximal region appears to be evolutionarily conserved. This proximal twist suggests that the A-C linker is able to adapt to the difference in angles between the microtubule triplets and thus remain connected to them.

## Discussion

In this study, we used cryo-ET to analyze the proximal region of centrioles from four evolutionarily distant species. We describe the structural features of this region, including the cartwheel, the D2-rod, the pinhead, the triplet base, and the A-C linker, which we integrate in a comprehensive model (Fig 6). Interestingly, we found that the cartwheel structure protrudes proximally beyond the microtubule triplets in all species that we investigated, especially in assembling *C. reinhardtii* procentrioles. These protruding cartwheel structures are complete, with spokes attached to the D2-rod. As this protrusion is also visible on the procentriole (Fig EV3), this observation supports the notion that the cartwheel assembles independently of the microtubule triplets, which are connected by the A-C linker. Although they can assemble independently, the cartwheel and A-C linker likely work synergistically to define the ninefold symmetry of the centriole (Nakazawa *et al*, 2007; Hilbert *et al*, 2016) as well as the cohesion of its proximal region (Yoshiba *et al*, 2019; Le Guennec *et al*, 2020). The cartwheel's proximal extension is also consistent with the proximal-directed growth of the cartwheel protein SAS-6 observed in *Drosophila* (Aydogan *et al*, 2018). It is currently not known whether the cartwheel structure can grow from its proximal end and whether such a mechanism is evolutionary conserved.

Our cryo-ET analysis revealed that the cartwheel's central hub in *C. reinhardtii* and *P. tetraurelia* is organized in ring pairs (Fig 6). Furthermore, in all four studied species, we observed densities inside the lumen of the central hub with a similar periodicity to the CIDs in *Trichonympha*. We therefore conclude that CID structures are present in every species studied to date and are a conserved element of the cartwheel. Moreover, one CID is positioned between the two rings of the ring pair, suggesting that it might be involved in ring pair assembly by helping build a cohesive unit. However, we could not

detect an asymmetric CID localization as reported by the accompanying Nazarov *et al* manuscript (Nazarov *et al*, 2020), probably due to our lower resolution and differences between species.

Concerning the molecular composition of the ring pair, it is possible that it consists of two stacked rings of SAS-6, but the distance of 3.1 nm observed in *P. tetraurelia* and 3.5 nm in *C. reinhardtii* does

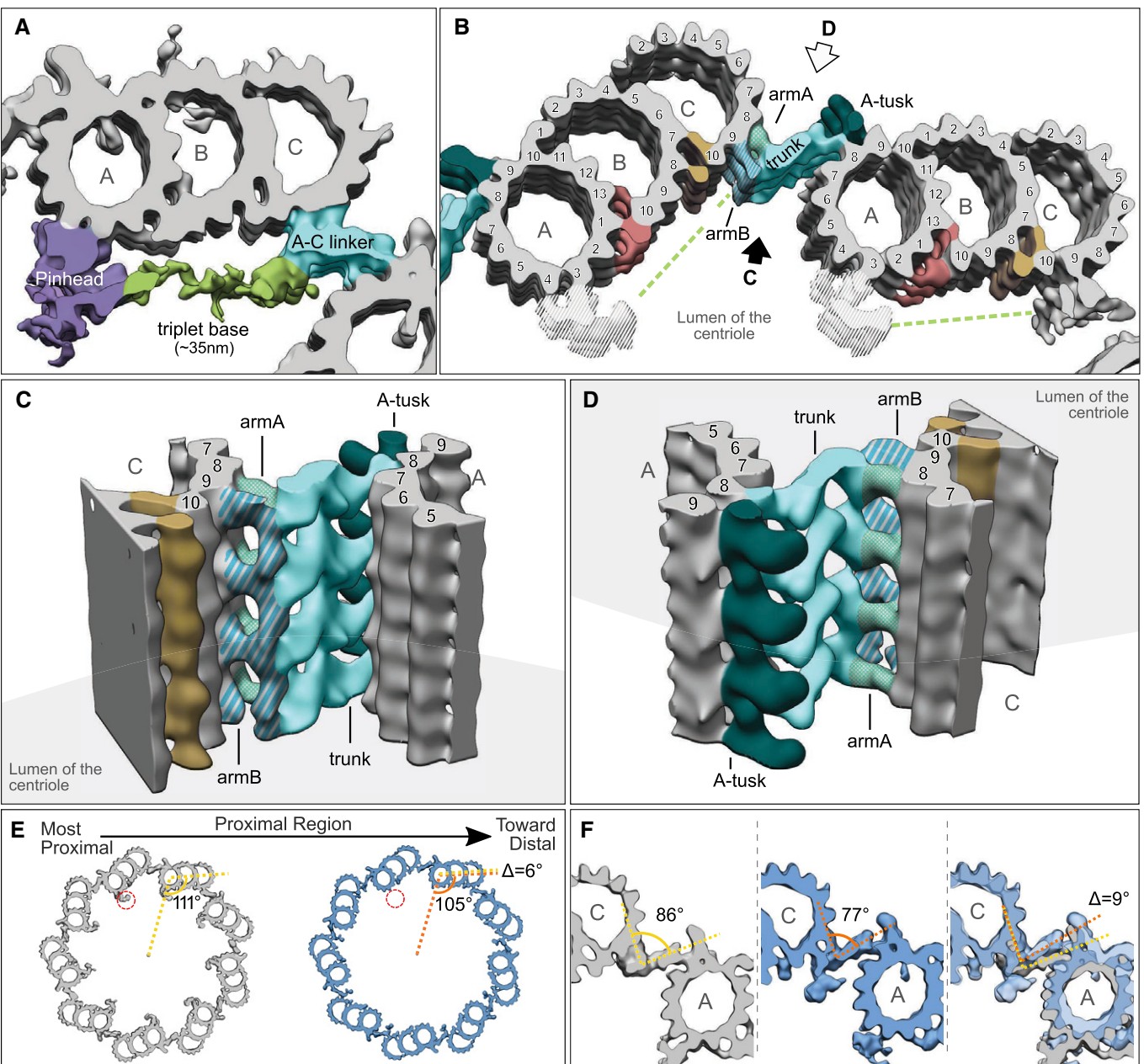

**Figure 5. Subtomogram averaging of the proximal triplet from *P. tetraurelia*.**

A   Microtubule triplet reconstruction from the beginning of the proximal region, displayed with a low contour threshold value to show the triplet base density (green) connected to the pinhead (purple) and the A-C linker (turquoise).

B   Two adjacent triplets from the beginning of proximal region, displayed with a higher contour threshold than in A. The A-C linker is segmented into different substructures (patterned turquoise colors) according to nomenclature (Li *et al*, 2019). The green dashed line indicates the putative position of the triplet base. Non-tubulin densities are colored in dark salmon and dijon. The pinhead has been hidden in this view, as its reconstruction is not correct due to the 8.5 nm initial subvolume picking that imposes this periodicity on the structure.

C   Three-dimensional side view of the A-C linker from the lumen of the centriole.

D   Three-dimensional side view of the A-C linker from outside the centriole (rotated 180° from C).

E   Top views of independent averages from the more proximal (gray) and more distal (blue) parts of the *P. tetraurelia* proximal region.

F   Focus on the A-C linker from the most proximal region (left, gray), the most distal proximal region (middle, blue), and the superimposition of both structures (right).

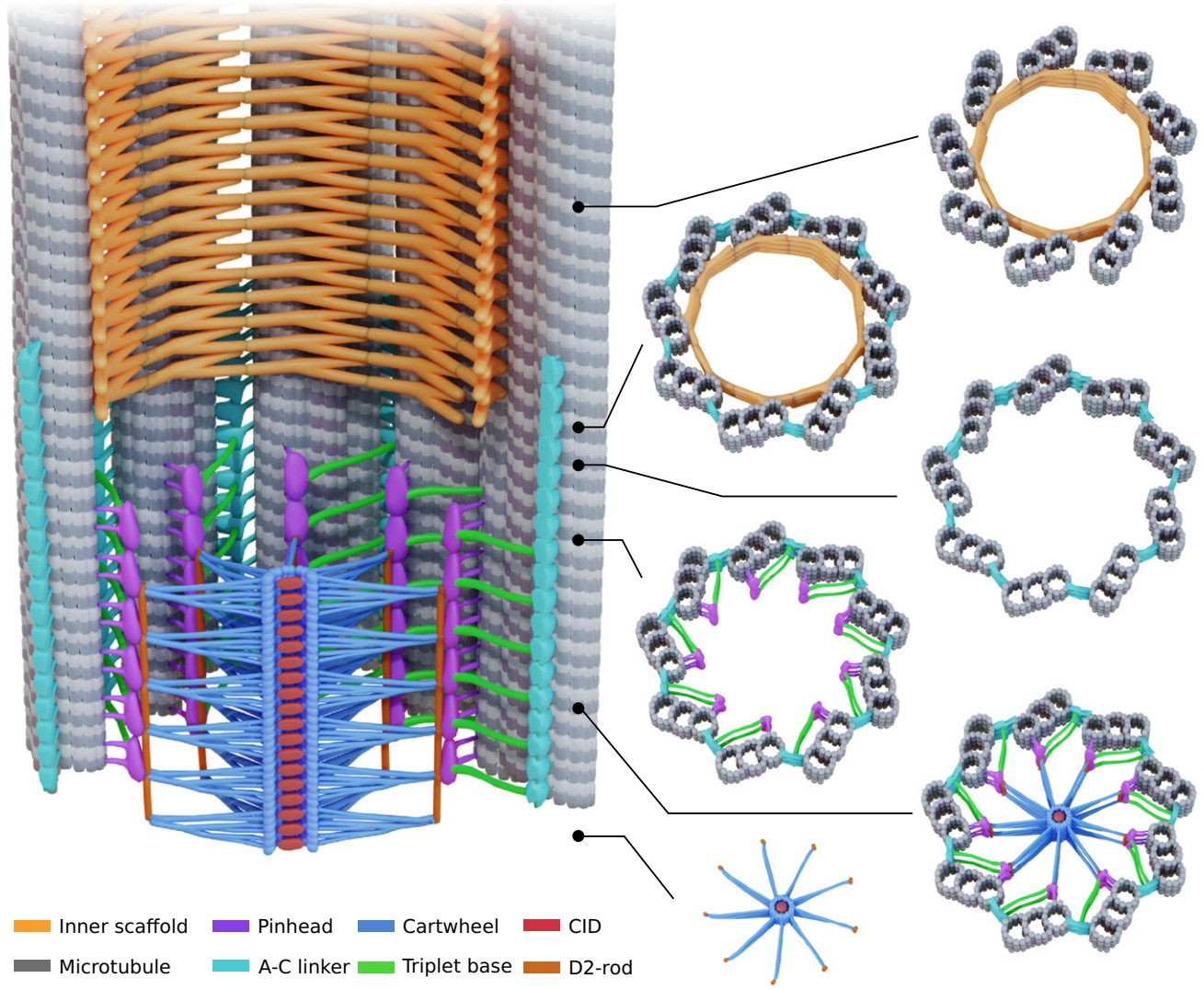

**Figure 6. Model of the architecture of the centriole's proximal region.**

The colors corresponding to each structure are indicated in the legend. Note that the cartwheel structure protrudes proximally from the microtubule wall; here, one unit has been depicted that corresponds to an external cartwheel of about 25 nm. The cartwheel's structural unit consists of 3 ring pairs, from which emanate 6 radial spokes that merge into a single D1 density before contacting the D2-rod adjacent to the pinhead structure. The pinhead and the A-C linker are connected through the triplet base. The A-C linker extends more distal than the cartwheel and co-exists with the inner scaffold structure.

not seem compatible with the 4.3 nm spacing observed in the ring stacking of the *L. major* SAS-6 crystallographic structure (van Breugel *et al*, 2014) (Appendix Fig S8A–C). However, similar to Nazarov *et al* (2020), we undertook fitting a LmSAS-6 ring pair into our best cryo-ET cartwheel map, that of *P. tetraurelia*. Surprisingly, we found that the LmSAS-6 ring crystallographic structure had a slightly smaller diameter than that of the *P. tetraurelia* hub (Appendix Fig S8D–F), making it impossible to fit without artificially stretching the SAS-6 proteins. This observation indicates that the size of the hubs can vary between species. In addition, as observed in Fig 3D, the emanating spokes from the rings are not in register but rather offset. This suggests that one SAS-6 ring would be rotated relative to another, allowing the SAS-6 subunits to slightly interdigitate, bringing the rings closer together by a few angstroms to obtain an inter-ring distance of 3.5 nm, as proposed by Nazarov *et al* (2020). However, the model proposed in *Trichonympha* is not entirely compatible with the *P. tetraurelia* ring offset we observed. In *P. tetraurelia*, the upper emanating spoke is positioned to the right of the one underneath, which is opposite to the orientation observed in *Trichonympha*. These discrepancies in hub diameter and ring offset might again reflect molecular divergences between species with different protein size and arrangement. Moreover, it is important to note that at our current resolution, it is difficult to ascertain whether the ring pair is composed solely of SAS-6, or contains an additional protein.

At the outer margin of the central hub's ring pairs, we observed that the cartwheel spokes are clearly organized differently than in *Trichonympha*, which turns out to be the biggest structural difference between the cartwheels of the different species. In *Trichonympha*, we could observe only two spokes merging, forming a longitudinal periodicity of 17 nm (Guichard *et al*, 2013; Nazarov *et al*, 2020). Here, we have demonstrated that the resolution obtained in the *Trichonympha* study is not sufficient to see certain details. Nevertheless, even by artificially lowering the resolution of our *P. tetraurelia* cartwheel map, the spoke organization remains distinct, with a lateral periodicity of ~25 nm. In both *C. reinhardtii* and *P. tetraurelia* cartwheels, this 25 nm periodicity results from the merger of spokes emanating from 3 adjacent ring pairs (Fig 6 and Appendix Fig S9). However, we could also distinguish that the spoke organization differed between these species. In *P. tetraurelia*, one spoke is made of 3 substructures that each emanate from a pair of rings, whereas in *C. reinhardtii*, the final spoke-tip is made from only two substructures (Fig 3J). As the coiled coil domain of SAS-6 is part of the spokes (Gönczy, 2012), the difference in radial spoke organization could potentially be explained by the low homology between SAS-6 coiled coils (Leidel *et al*, 2005). It is possible to imagine that coiled coils of neighboring SAS-6 proteins merge to form a coiled coil bundle or a tetramer/hexamer. Another possibility is that a different protein interacts with the SAS-6 coiled coil and is responsible for this bundling. To date, SAS-5 is one of the most likely candidates for this role. Indeed, it has been shown in several species that SAS-5 interacts with the SAS-6 coiled coil where the bundle is formed (Qiao *et al*, 2012; Cottee *et al*, 2013; Shimanovskaya *et al*, 2013). In addition, it has been shown that the Ana2 (SAS-5 in *Drosophila*) coiled coil forms a tetramer (Cottee *et al*, 2013) and that *C. elegans* SAS-5 forms higher-order protein assemblies up to hexamers in solution (Rogala *et al*, 2015). It is therefore possible that different stoichiometries of SAS-6:SAS-5 can modify the architecture of the spoke bundling.

Our study also highlights the triplet base structure (Fig 6), originally described in conventional electron microscopy of resin-embedded mammalian centrioles (Vorobjev & Chentsov, 1980). We found that the triplet base connects the pinhead to the A-C linker, thus forming a continuous structure that bridges the cartwheel with the A-C linker. The triplet base might enhance the cohesion and stability of the entire proximal region. Although its molecular nature is not known, its apparent flexibility, length, and low map density, similar to the cartwheel spokes, would suggest that the triplet base is made by a long coiled coil protein. It is therefore tempting to speculate that this structure might consist of the coiled coil protein Bld10p/Cep135. Indeed, based on its immuno-localization as well as its known interaction with the C-terminus of SAS-6 and microtubules, current models place this protein as part of the pinhead (Hiraki *et al*, 2007; Hirono, 2014; Kraatz *et al*, 2016). The coiled coil length prediction for Cep135 is ~900 of its 1140 total amino acids, which would yield a coiled coil that is 133 nm long (900 residues × 0.1485 nm [axial rise per residue] = 133 nm, formula from (Kitagawa *et al*, 2011)). Considering that the pinhead is ~20 nm long (Guichard *et al*, 2013), it is likely that a large portion of Bld10/Cep135 extends from it. Therefore, we hypothesize that a part of the predicted 133 nm coiled coil constitutes the 35 nm long triplet base connecting to the A-C linker (Fig 5A). This hypothesis is consistent with the phenotypes of *C. reinhardtii* and *Tetrahymena*

Bld10p mutants, which not only lose the connection of the cartwheel to the microtubule wall but also lose the microtubule triplets themselves, suggesting that the cohesion between triplets is partially lost (Matsuura *et al*, 2004; Bayless *et al*, 2012). Future studies on the precise location of the different regions of Cep135 would be needed to answer these questions.

An important structural feature revealed in our study is the intrinsic polarity of the cartwheel along its proximal–distal axis. Previous work had observed such polarity in the pinhead and A-C linker structures (Guichard *et al*, 2013; Li *et al*, 2019). Our work now reveals that polarity also exists within the cartwheel itself, which might play a critical role in centriole biogenesis. Such polarity is likely important to define the directionality of structural features that assemble after cartwheel formation. For instance, microtubule triplets, which are also polarized structures, only grow in the distal direction. Although it is possible that the triplets slightly lengthen on the proximal side, it is clear that the plus ends of the microtubules always face towards the distal end of the centriole. It is therefore possible that the polarity of the cartwheel defines the growth directionality of the procentriole from the very beginning of assembly. It is interesting to note that the only known example of microtubule triplet polarity inversion was observed in a *Tetrahymena* Bld10p mutant (Bayless *et al*, 2012). As Bld10p constitutes part of the cartwheel spoke-tip/pinhead, this reinforces the idea that the cartwheel defines the direction of centriole growth.

Combining our present study with previous work on the structure of the centriole proximal region from different species offers a glimpse at evolutionary conservation and divergence at the level of molecular architecture. The data presented here suggest that the cartwheel-containing region has a conserved overall organization with defined structural characteristics (Fig 6). However, our work also demonstrates that the specific layout of the centriole and the finer structural elements may differ considerably between species. These observations correlate well with the fact that many centriolar proteins are conserved between species, yet they can vary significantly in their size or amino acid composition, as exemplified by the low sequence homology of the cartwheel protein SAS-5/Ana2/STIL (Stevens *et al*, 2010). Our work therefore shows that there may be different routes to build a centriole.

## Materials and Methods

### *Paramecium tetraurelia* centriole isolation and cryo-electron tomography

*P. tetraurelia* cortical units were isolated from two different strains, the wild-type reference strain d4-2 and Δ-CenBP1, as previously described (Le Guennec *et al*, 2020). Briefly, isolated *P. tetraurelia* centrioles were diluted with 1:1 colloidal gold in 10 mM K-PIPES buffer. Five microliters were deposited on 300 mesh lacey carbon grid and blotted from the backside before plunging in liquid ethane using a manual plunge freezing system. Tomograms were acquired with SerialEM software (Mastronarde, 2005) on a 300 kV FEI Titan Krios equipped with a Gatan K2 summit direct electron detector. The tilt series were recorded from approximately −60° to +60° (bidirectional, 2° steps, separated at −0°), using an object pixel size of 3.45 Å, a defocus around −5 μm and a total dose of 70–120 electrons/Å$^2$.

## Culture and *in situ* tomography of *Chlamydomonas reinhardtii* cells

The *in situ* of FIB-milling of *C. reinhardtii* centrioles was performed in the *mat3-4* strain, as previously described (Le Guennec *et al*, 2020). In brief, 4 μl of *C. reinhardtii* cells was deposited onto 200-mesh copper EM (R2/1, Quantifoil Micro Tools) and vitrified using a Vitrobot Mark 4 (FEI Thermo Fisher Scientific). Cryo-FIB sample preparation was performed as previously described (Schaffer *et al*, 2015, 2017). The FIB-milled EM grids were transferred into a 300-kV FEI Titan Krios transmission electron microscope, equipped with a post-column energy filter (Quantum, Gatan) and a direct detector camera (K2 Summit, Gatan). Tomogram were acquired using SerialEM software (Mastronarde, 2005), with tilt series between −60° and +60° (bidirectional, 2° steps, separated at −0° or −20°) and a total dose around 100 electrons/Å$^2$. A subset of tilt series was acquired with a dose-symmetric scheme (Hagen *et al*, 2017). Individual tilts were recorded in movie mode at 12 frames/s, at an object pixel size of 3.42 Å and a defocus of −5 to −6 μm.

## *Naegleria gruberi* centriole isolation and cryo-electron tomography

Centriole isolation and tomogram acquisition were performed as previously described in (Le Guennec *et al*, 2020). Briefly, the *N. gruberi* NEG strain was differentiated into flagellates (Fulton, 1977), and centrioles were isolated using a sucrose gradient. Isolated centrioles were then deposited onto 200-mesh copper EM grids coated with holey carbon (R3.5/1, Quantifoil Micro Tools) and plunge-frozen in a liquid ethane/propane mixture. Tilt series were recorded using SerialEM (Mastronarde, 2005) on a 300 kV FEI Titan Krios transmission electron microscope, equipped with a direct detector camera (K2 Summit, Gatan) and a post-column energy filter (Quantum, Gatan). Tilt series were bidirectional (2° steps, separated at −0° or −20°), and individual images were recorded in movie mode at 10 frames/s, with an object pixel size of 4.21 Å and a defocus of −5 to −8 μm.

## Human centriole isolation and cryo-electron tomography

Human centrioles were isolated from the human lymphoblastic KE-37 cell line as previously described (Gogendeau *et al*, 2015), with modification described in (Le Guennec *et al*, 2020). In brief, 5 μl of isolated centrioles diluted 1:2 with colloidal gold in 10 mM K-PIPES buffer was deposited on 300 mesh lacey carbon grids, blotted from the backside and quickly vitrified in liquid ethane using a manual plunge freezing. Tomogram acquisition was performed a 300 kV FEI Titan Krios equipped with a Gatan K2 summit direct electron detector. Bidirectional tilt series (2° steps, separated at −20°) were acquired with SerialEM (Mastronarde, 2005). Each tilt was recorded in movie mode at 12 frames/s with an object pixel size of 3.42 Å and a defocus of −4 to −6 μm.

## Radial spoke periodicity extraction

To identify the periodicity of the spokes, we performed a translational analysis, as previously used to determine the cartwheel periodicity in *Trichonympha* (Guichard *et al*, 2012). A cross section with a thickness of 143 nm was extracted from the proximal regions of *P. tetraurelia* and *C. reinhardtii* centrioles (Appendix Fig S9). To increase the contrast, volumes were binned by a factor of 2, then symmetrized, and filtered using a 3D Gaussian filter with Fiji (Schindelin *et al*, 2012). To avoid signal from the microtubule triplets and the cartwheel, we applied a mask to conserve only signal coming from the radial spokes. Using SPIDER (Frank *et al*, 1996), volumes were shifted along the *z*-axis every pixel from −59 to +59 (corresponding to −40 to +40 nm). For each translation, the volume obtained was compared to the non-shifted original volume by calculating the cross correlation.

## Subtomogram averaging of the cartwheel

### *P. tetraurelia* cartwheel

From 7 tomograms, 10 intact cartwheels were extracted as subtomograms with dimensions of 420 × 420 × 420 voxels. For each cartwheel, 9 duplicates were generated, and each of them was rotated by a multiple of 40° to produce 9 different orientations of the original cartwheel. Each new volume was then shifted by −25, 0, or +25 nm to position a different unit of the cartwheel in the center of the volume. For each cartwheel, 27 subtomograms were generated (9 orientations × 3 units), resulting in 270 subtomograms in total from 10 cartwheels. To reduce the noise, the subtomograms were filtered using the non-linear anisotropic diffusion command of Bsoft (Heymann *et al*, 2008).

An initial reference was generated by taking a cartwheel and its 8 differently oriented copies and averaging them together. The 270 subtomograms were aligned on this reference using SPIDER (Frank *et al*, 1996). After a few iterations, the average generated was used as a new reference on which the original, filtered but not aligned, subtomograms were aligned. From the 270 subtomograms, 38 failed to correctly align and thus were removed from the final set, resulting in 232 subtomograms used for the averaging. Ninefold symmetry was then applied on the generated map to increase the contrast of the volume.

### *C. reinhardtii* cartwheel

From five bin2 tomograms, 5 cartwheels were extracted as subtomograms with dimensions of 210 × 210 × 210 voxels. For each cartwheel, 9 duplicates were generated, and each duplicate was rotated by a multiple of 40° to generate 9 different orientations of the original cartwheel. Each rotated volume was then shifted by × 25, 0, or +25 nm to position different units of the cartwheel in the center of the volume. From five cartwheels, 9 × 3 = 27 subtomograms were generated resulting in 135 subtomograms in total. To improve the contrast, subtomograms were binned by a factor 2.

The 135 subtomograms were first aligned on the *P. tetraurelia* cartwheel map previously generated. Out of the 135 subtomograms, 86 were correctly aligned and used to produce an average map. This map was filtered by applying 3 iterations of Gaussian filter (with a sigma value of 2). The originally unaligned subtomograms were then aligned on this filtered average. 102 subtomograms were correctly aligned and kept to generate the average map. Ninefold symmetry was then applied on the generated map to increase the contrast of the volume.

## Subtomogram averaging of the A-C linker

From 11 tomograms of *P. tetraurelia* centrioles, 16 centrioles contained an intact proximal region. The positions of microtubules triplets were picked and interpolated every 8.5 nm as described in Le Guennec *et al* (2020) along the region displaying the A-C linker structure. Using Dynamo (Castaño-Díez *et al*, 2012), 1941 subtomograms of 320 × 320 × 320 voxels were extracted, encompassing the microtubule triplet with its associated A-C linkers. Initially, the microtubule triplets were roughly aligned to the *Trichonympha* reference (EMD-2330) (Guichard *et al*, 2013). To discriminate between subtomograms from the most proximal- and the most distal regions, a mask was created around the A-B inner junction where either the pinhead (a proximal marker) or the inner scaffold (a more distal marker) lies. Multireference alignment was performed on this region, allowing us to classify our set into two classes: the "most proximal" class (*n* = 1,042) and the "most distal" class (*n* = 899), as depicted in Appendix Fig S7. For each set, the average was generated as a reference for the next alignment step. Each set was then divided into two independent halves and aligned for a few iterations to produce two averages. The resolution was estimated by generating the Fourier shell correlation (FSC) curve from the 2 averages using the EMAN2 package and choosing a cutoff at 0.143 (Appendix Fig S6). One of the averages was bandpass filtered at this resolution, and the two half-sets were aligned on this filtered map to generate the final map.

The new aligned set was then split again into two halves; each half was locally aligned on the A-C linker region of the final map. After the two halves were aligned and the resolution computed, they were aligned on a common filtered map as previously performed for the global map.

The global map and the A-C linker map were combined together as described in Le Guennec *et al* (2020) to generate a volume displaying two adjacent microtubule triplets connected through the A-C linker. This map was then binned by a factor of 2 and combined with a rotated duplicate of itself to form a structure of the complete ninefold proximal region, as described in Le Guennec *et al* (2020).

## Symmetrization

Top views of centrioles were generated using a z-projection of few slices from the cryo-tomogram and processed with the ImageJ plugin CentrioleJ for circularization and symmetrization (Guichard *et al*, 2013).

The symmetrization of the CID region was performed by generating a z-projection of a proximal part centered on the CID. From this image, 9 duplicates were generated by applying rotation from 0 to 360 degrees with a step of 40 degrees using Bsoft (Heymann *et al*, 2008). The 9 rotated images were then averaged together using SPIDER (Frank *et al*, 1996).

Similarly, for the cartwheel protrusion, 9 duplicates of the volumes were generated, rotated and averaged together to create the symmetrized cartwheel protrusion region.

## Transmission electron microscopy of *Naegleria gruberi* serial section

*N. gruberi* NEG cells were differentiated from amoebae into flagellates as described in Le Guennec *et al* (2020), following a standard protocol (Fulton, 1977). Cells were fixed 50–80 min after the initiation of differentiation in order to observe both procentrioles and mature centrioles. The cells were pelleted and resuspended in 60 mM HEPES, 4 mM $CaCl_2$, 2.5% glutaraldehyde, pH 7.2, and fixed for 120 min at room temp (replacing the fixative with fresh solution after 40 min). Cells were washed 2 × 5 min in 60 mM HEPES, 4 mM $CaCl_2$, pH 7.2, and osmicated using 1% $OsO_4$ in distilled water for 75 min at 4°C. Cells were washed 3 × 10 min in distilled water before en bloc staining in 1% uranyl acetate in distilled water overnight at 4°C. After washing 3 × 10 min in distilled water, the cells were embedded in 1% Agar noble (BD Difco, Sparks, MD, USA). Dehydration in ethanol, infiltration with Epon 812 (Serva Electrophoresis, Heidelberg, Germany), and final embedding were performed following standard procedures. Ultrathin serial sections (nominal 60 nm thickness) were cut with a diamond knife (type ultra 35°; Diatome, Biel, Switzerland) on an EM UC6 ultramicrotome (Leica, Wetzlar, Germany) and mounted on single-slot Pioloform-coated copper grids (Plano, Wetzlar, Germany). Sections were stained with uranyl acetate and lead citrate (Reynolds, 1963) and viewed with a JEM-2100 transmission electron microscope (JEOL, Tokyo, Japan) operated at 80 kV. Micrographs were acquired using a 4K charge-coupled device camera (UltraScan 4000; Gatan, Pleasanton, CA) and Gatan Digital Micrograph software (version 1.70.16.).

## Data availability

Subtomogram averages have been deposited at the Electron Microscopy Data Bank (https://www.ebi.ac.uk/pdbe/emdb/) with the accession codes EMD-10726, EMD-10727, EMD-10728, EMD-10729. Correspondence and requests for materials should be addressed to P.G. (paul.guichard@unige.ch).

*Expanded View* for this article is available online.

## Acknowledgements

We thank the BioImaging Center at Unige. We thank Jürgen Plitzko and Wolfgang Baumeister for providing support and instrumentation. We thank Chandler Fulton and Lillian Fritz-Laylin for providing *N. gruberi* cultures and advice. This work was supported by the Swiss National Science Foundation (SNSF) PP00P3_187198 and by the European Research Council ERC ACCENT StG 715289 attributed to P.G., as well as the Helmholtz Zentrum München and the Max Plank Society. Open access funding enabled and organized by Projekt DEAL.

## Author contributions

VH, PG, and BDE conceived, supervised, designed the project, and wrote the final manuscript with input from all authors. NK and MLG performed all image processing and analyzed the data. A-MT purified the *P. tetraurelia* centrioles. NK isolated the human centrioles and acquired tomograms of these two species with the help of LK, KNG, HS, HvdH, and BDE Sample preparation and tomography of *in situ C. reinhardtii* centrioles and isolated *N. gruberi* centrioles was performed by PSE, MS, HvdH, and BDE. GA generated the 3D model of the centriole. SG performed the electron microscopy of *N. gruberi* centrioles. YS contributed to Appendix Fig S8.

## Conflict of interest

The authors declare that they have no conflict of interest.

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
