## [Review Process File · The EMBO Journal]

Architecture of the centriole cartwheel-containing region revealed by cryo-electron tomography

Nikolai Klena, Maeva Le Guennec, Anne-Marie Tassin, Hugo Van den Hoek, Philipp Erdmann, Miroslava Schaffer, Stefan Geimer, Gabriel Aeschlimann, Lubomir Kovacic, Yashar Sadian, Kenneth Goldie, Henning Stahlberg, Benjamin Engel, Virginie Hamel, and Paul Guichard
DOI: 10.15252/embj.2020106246

Corresponding author(s): Paul Guichard (paul.guichard@unige.ch)

Review Timeline:

Transfer from Review Commons:	15th Jul 20
Editorial Decision:	31st Jul 20
Revision Received:	20th Aug 20
Accepted:	24th Aug 20

Editor: Hartmut Vodermaier

Transaction Report: This manuscript was transferred to The EMBO Journal following peer review at Review Commons.

**Review
COMMONS**

Reviewer #1

Structural study on the centriolar cartwheel has been limited to exceptional flagellate *Trichonympha*, while the microtubule triplet of this region was investigated by David Agard's group (*Chlamydomonas* and mammalian; Li et al. 2019 and Greenan et al. 2020). This work, Klena et al. analyzed 3D structure of the cartwheels from *Chlamydomonas*, *Paramecium*, *Naegleria* and human. Structural analysis of cartwheels has been a challenging topic, due to short length (differently from exceptionally long *Trichonympha*) and thus poor signal-to-noise ratio during image analysis. They employed high-end technique such as cryo-FIB milling, 300kV TEM, and direct electron detection to maximize the quality of the tomograms and conducted subtomogram averaging. Using cryo-ET and subtomogram averaging, they provided beautiful geometrical description of the central hub of cartwheel, which is made of the SAS-6 protein, one key of nine-fold symmetry, as well as the joint between the cartwheel and the triplet microtubule from *Paramecium* and *Chlamydomonas* and discussed structural differences between these two species. Regarding *Naegleria* and human, they compare length of the cartwheel and areas of the triplet decorated by different structural features.

This reviewer would recommend the manuscript for publication, once 3D structure of the cartwheel central hub and spokes from *Naegleria* and human are reconstructed by subtomogram averaging for further comparative structural studies and several technical issues mentioned below are clarified. This paper after addressing these points will attract the wide readers' attention.

Major points

1. Subtomogram averaging of human and *Naegleria* cartwheels and triplets

The authors stress evolutionary insights obtained in this work (Line 471). Subtomogram averaging of human and *Naegleria* centrioles, both for the central hub and the triplet at the cartwheel region is necessary. Especially human cartwheel structure is awaited and thus will make this work worth for publication in a Journal of high reputation. Is the periodicity and conformation of the hub and the spoke same as one of the three unicellular organisms? Do they find any novel structural feature in addition to Greenan et al. reported? How about structure inside the hub?

2. Periodicity of the cartwheel

The authors conducted subtomogram averaging by picking particles with 25nm distance (Line843, Line860). This will lead the following analysis to the 25nm periodicity assumed and may cause a risk of structure not following 25nm periodicity being smeared out. Therefore longitudinal periodicity must be carefully confirmed. Fourier analysis (such as Fig.S3CF of the other manuscript, Nazarov et al.) is one way to prove periodicity in unbiased way, but might be difficult with short cartwheels. This reviewer would propose classification of randomly extracted (but of course along the cylindrical hub or along the triplet microtubules) subtomograms. This experiment will end up with multiple subaverages, which are 25nm (or multiple times of that) shifted from each other. Then it will prove their assumption. The alternative way is to average assuming long enough periodicity, which is the least common multiple of all the possible periodicity. This reviewer is not optimistic, since the cartwheels studied in this work is relatively short.

The authors also need to clarify if the periodicities of the central hub from *Chlamydomonas* in this work (4.0nm in Line135), their original paper (8.5nm), CID (8.7nm in Line142) and the microtubule triplet correlate each other or are independent. If they are independent, heterogeneity will come from this periodicity mismatch. Image classification of parts with masking other part of the volume will clarify this.

3. Classification

They employed multi-reference image classification (Line 820-822) to detect polymorphism inside the cartwheel, assuming that there are two conformations coexisting and there are two conformations depending on the proximity. What is the basis of this classification strategy? What kind of volume was used as a template? The distribution of the multiple structures should be shown (like Fig.S5F from Nazarov et al. 2020, another paper of them).

4. Quality of Naegleria and human tomograms

Quality of tomograms from *N. gruberi* (Fig.2B, in which the central hub is broken) and human (Fig.2C) is not good enough to discuss cartwheel geometry in a convincing way. This reviewer imagines that cartwheels were damaged during preparation from *N. gruberi* (as happened in Li et al. 2012 and improved by them Guichard et al. 2013). Preparation with shorter time will help. Fig. 2B (and other images with similar quality) must be replaced by better images and data from this tomogram should be excluded from the statistics in Fig.2J and others.

5. Quality of subtomogram averaging

Image quality of the averaged subtomograms is not so high as expected from FSC-based resolution estimation (similar to the resolution of Li et al. 2019), judging from appearance (separation of adjacent tubulins) of the doublet microtubules. What is the reason? Can their way to calculate resolution with 0.143 cut-off be considered

Structural study on the centriolar cartwheel has been limited to exceptional flagellate *Trichonympha*, while the microtubule triplet of this region was investigated by David Agard's group (*Chlamydomonas* and mammalian; Li et al. 2019 and Greenan et al. 2020). This work, Klena et al. analyzed 3D structure of the cartwheels from *Chlamydomonas*, *Paramecium*, *Naegleria* and human. Structural analysis of cartwheels has been a challenging topic, due to short length (differently from exceptionally long *Trichonympha*) and thus poor signal-to-noise ratio during image analysis. They employed high-end technique such as cryo-FIB milling, 300kV TEM, and direct electron detection to maximize the quality of the tomograms and conducted subtomogram averaging. Using cryo-ET and subtomogram averaging, they provided beautiful geometrical description of the central hub of cartwheel, which is made of the SAS-6 protein, one key of nine-fold symmetry, as well as the joint between the cartwheel and the triplet microtubule from *Paramecium* and *Chlamydomonas* and discussed structural differences between these two species. Regarding *Naegleria* and human, they compare length of the cartwheel and areas of the triplet decorated by different structural features.

This reviewer would recommend the manuscript for publication, once 3D structure of the cartwheel central hub and spokes from *Naegleria* and human are reconstructed by subtomogram averaging for further comparative structural studies and several technical issues mentioned below are clarified. This paper after addressing these points will attract the wide readers' attention.

****Major points****

1. Subtomogram averaging of human and *Naegleria* cartwheels and triplets

The authors stress evolutionary insights obtained in this work (Line 471). Subtomogram averaging of human and *Naegleria* centrioles, both for the central hub and the triplet at the cartwheel region is necessary. Especially human cartwheel structure is awaited and thus will make this work worth for publication in a Journal of high reputation. Is the periodicity and conformation of the hub and the spoke same as one of the three unicellular organisms? Do they find any novel structural feature in addition to Greenan et al. reported? How about structure inside the hub?

2. Periodicity of the cartwheel

The authors conducted subtomogram averaging by picking particles with 25nm distance (Line843, Line860). This will lead the following analysis to the 25nm periodicity assumed and may cause a risk of structure not following 25nm periodicity being smeared out. Therefore longitudinal periodicity must be carefully confirmed. Fourier analysis (such as Fig.S3CF of the other manuscript, Nazarov et al.) is one way to prove periodicity in unbiased way, but might be difficult with short cartwheels. This reviewer would propose classification of randomly extracted (but of course along the cylindrical hub or along the triplet microtubules) subtomograms. This experiment will end up with multiple subaverages, which are 25nm (or multiple times of that) shifted from each other. Then it will prove their assumption. The alternative way is to average assuming long enough periodicity, which is the least common multiple of all the possible periodicity. This reviewer is not optimistic, since the cartwheels studied in this work is relatively short.

The authors also need to clarify if the periodicities of the central hub from *Chlamydomonas* in this work (4.0nm in Line135), their original paper (8.5nm), CID (8.7nm in Line142) and the microtubule triplet correlate each other or are independent. If they are independent, heterogeneity will come from this periodicity mismatch. Image classification of parts with masking other part of the volume will clarify this.

3. Classification

They employed multi-reference image classification (Line820-822) to detect polymorphism inside the cartwheel, assuming that there are two conformations coexisting and there are two conformations depending on the proximity. What is the basis of this classification strategy? What kind of volume was used as a template? The distribution of the multiple structures should be shown (like Fig.S5F from Nazarov et al. 2020, another paper of them).

4. Quality of Naegleria and human tomograms

Quality of tomograms from *N. gruberi* (Fig.2B, in which the central hub is broken) and human (Fig.2C) is not good enough to discuss cartwheel geometry in a convincing way. This reviewer imagines that cartwheels were damaged during preparation from *N. gruberi* (as happened in Li et al. 2012 and improved by them Guichard et al. 2013). Preparation with shorter time will help. Fig. 2B (and other images with similar quality) must be replaced by better images and data from this tomogram should be excluded from the statistics in Fig.2J and others.

5. Quality of subtomogram averaging

Image quality of the averaged subtomograms is not so high as expected from FSC-based resolution estimation (similar to the resolution of Li et al. 2019), judging from appearance (separation of adjacent tubulins) of the doublet microtubules. What is the reason? Can their way to calculate resolution with 0.143 cut-off be considered as gold standard FSC? For gold standard FSC, once separated subtomograms must be aligned, averaged and further iterated without cross-talk. After extraction by Dynamo, which program did they use for subtomogram analysis (the other paper used Relion, thus gold standard FSC was properly calculated, this reviewer believes)?

6. Cartwheel protrusion

Does the proximal cartwheel protrusion (Fig. 2KL) have similar spoke structure as the main part of the cartwheel? They modeled as if the cartwheel spoke has the same conformation there as in the main part of the cartwheel (Fig.6), but without experimental basis. By averaging subtomograms from the cartwheel, the authors will see if the conformation of the spoke is

determined by the triplet microtubule binding or stable by itself.

****Minor points:****

Line 233: green dotted lines do not exist in Fig.3DJ. There should be more visible way to present "slightly tilted spokes", for example by a movie.

Line 237: Periodic features are visible in the right panels of Fig.S6AB, but they seem not to be raw data, but after some operation (not written in the legend; please clarify). The central panels are the raw data, but periodic feature is not clear in the top center panel.

Line 440: triplet base

They attempted to interpret molecular identification of the triplet base. It is fine to discuss/model, but too detailed discussion only based on molecular weight and coiled-coil prediction will not add any reliability to this model, since the composition of this area is not known. To locate proteins, either deletion mutant analysis, label (such as antibody) or genetic tagging, or biochemical methods (such as cross-linking MS) is necessary. This reviewer recommends removal of too detailed statement (Line 442-449).

Line 555 (Fig.3AB): it is confusing to use yellow dotted line for two different objects in Fig.3A and B. it would be more convenient to use different colors.

Fig.3I: a similar surface rendering presentation should be shown for Chlamydomonas data as well.

Significance

This reviewer would recommend the manuscript for publication, once 3D structure of the cartwheel central hub and spokes from Naegleria and human are reconstructed by subtomogram averaging for further comparative structural studies and several technical issues mentioned below are clarified. This paper after addressing these points will attract the wide readers' attention.

Reviewer #2

In this manuscript Klena et al., use cryo-electron tomography (CET) to analyse the structure of the proximal end of centrioles and procentrioles. These tiny organelles have many important functions in cells so there is considerable interest in understanding how they are assembled and function. While there has been great progress in defining the proteins required for centriole assembly/function, their small size has meant we have very limited information about their detailed structure. The Hachet/Guichard team have been applying CET methods to this problem for some time (initially with Pierre Gonczy), and in this manuscript they use the most up-to-date technology to analyse centriole structure in several species-comparing this to the lower resolution structures from Trichonympha they published previously (Guichard et al., Science, 2012; Guichard et al., Curr. Biol., 2013).

I am not an expert in CET, so it will be important to have a review from someone with this technical knowledge (although this team are real experts, so I would not anticipate any problems). As a biologist, the data looks very impressive; really beautiful images and a compelling narrative that was generally well presented and easy to follow. I found this paper fascinating, and these structures provide several important insights that will profoundly influence our thinking on how these organelles assemble and function. Several aspects of the overall architecture are now shown to be widely conserved-such as the extension of the cartwheel below

the centriole MTs, the CID, and the architecture of the linkage between the cartwheel and the MTs (although several groups have recently published quite similar data focused on the architecture of the centriole MTs). Equally important are those aspects that seem to vary between species, most surprisingly the organisation of the cartwheel spokes that link the inner hub to the outer MTs. These findings will be of great interest to the centriole and centrosome fields and I strongly support publication in The EMBO J. I have only a few points that the authors should consider.

1. Perhaps the most important is the relationship of this paper to the Nazarov et al., paper that was co-submitted by the Gonczy laboratory. The two papers complement each other well, and I strongly support back-to-back publication. However, the papers come to slightly different conclusions on a number of points, and it would be very useful to the field if these could be addressed/discussed so we can hear the author's thoughts on these points.

1A. The first issue is cartwheel polarity. The authors do not make much of polarity here, but they conclude that the spokes emanating from the hub are asymmetric (Fig. 3I). They say this is true for both *P. tetaurelia* and *C. reinhardtii*, but only show this for the former and it looks much less clear for the latter-so this should be shown properly, or at least discussed in more depth. On the basis of this they conclude that the core 25nm repetitive unit of the cartwheel has intrinsic polarity. Nazarov et al., make the same conclusion, but base this on the asymmetric localisation of the CID within the hub, something that is not reported here. This may be due to species differences, but it is important to know whether these authors have looked for this and what they conclude.

1B. The second issue is that Nazarov et al., provide evidence that strongly suggests that in some species the double stacked Sas-6 rings are offset from one another. Again, it would be important to know if Klena et al. have looked for this? Perhaps the staggering of the 3 spokes in the *P. tetaurelia* structure could be interpreted in this way (Figure 3J)?

1C. Finally, I was surprised that the authors did not try to model the crystal structure of Sas-6 into their various cartwheel models, especially in *C. reinhardtii* where the Sas-6 structure is known (but also for the other species using the *C. reinhardtii* structure, which is highly conserved). It perhaps seems obvious how the Sas-6 rings might be arranged in these structures, but this overall organisation of the spokes is quite surprising (and quite different to how originally envisaged in the more regular *Triconympha* structure). Nazarov et al., perform this sort of analysis, and the Sas-6 structure fits as one might have expected, but this analysis provides some support for the offsetting of the stacked rings and I would be very interested to see it here.

2. To the non-expert eye it is quite difficult to understand how meaningful data can be extracted from images that look like the human cartwheel presented in Fig.2C. I don't doubt that this is the case, but it might be worth briefly describing how this was done in the legend (or pointing to a section in the Materials and Methods if a slightly longer explanation is required). One can see that the human data is more variable, but it would be good to know how the authors decided what was suitable for analysis (perhaps, for example, only the CID highlighted with an arrow was counted in this image?).

Significance

See above.

Reviewer #3

****Summary****

In this manuscript, Klena and colleagues describe the high-resolution architecture of centrioles from four different species using cryo-electron tomography. They mainly focus on the cartwheel-containing region which is important for early events of centriole duplication. The authors revealed several conserved features in four different species (*Chlamydomonas reinhardtii*, *Paramecium tetraurelia*, *Naegleria gruberi*, and human). For example, the authors show that the periodicity of the cartwheel central hub is about ~4 nm, which was previously shown to be 8.5 nm from *Trichonympha*. They also determined the cartwheel inner density (CID) and proximal protrusion of cartwheel in all investigated species. The authors should consider the following:

****Major point****

Line 263 to 276: To solve the discrepancy in central hub periodicity between *Trichonympha* and *P. tetraurelia*, the authors lowered the resolution of images from *P. tetraurelia*. However, it would be better to show high resolution images of *Trichonympha* to support this possibility.

Significance

Their experimental results are of very high quality and provide clues as to which organizational features of their architectures are conserved or not. I believe this manuscript would be interesting for anyone who is researching about the centrosomes. Therefore, I would like to recommend this paper to be published in the EMBO journal which is affiliated with the Review Commons.

Review Commons Referred Preprint RC-2020-00252

We would like to thank the editors and reviewers for their comments and suggestions to further improve the quality of our manuscript. In this letter, we provide answers and details addressing the aforementioned points. Note that the additional text is highlighted in red and the referred sentences are underlined to ease following-up the modifications within the revised manuscript.

Reviewer #1 (Evidence, reproducibility and clarity (Required)):

Structural study on the centriolar cartwheel has been limited to exceptional flagellate *Trichonympha*, while the microtubule triplet of this region was investigated by David Agard's group (*Chlamydomonas* and mammalian; Li et al. 2019 and Greenan et al. 2020). This work, Klena et al. analyzed 3D structure of the cartwheels from *Chlamydomonas*, *Paramecium*, *Naegleria* and human. Structural analysis of cartwheels has been a challenging topic, due to short length (differently from exceptionally long *Trichonympha*) and thus poor signal-to-noise ratio during image analysis. They employed high-end technique such as cryo-FIB milling, 300kV TEM, and direct electron detection to maximize the quality of the tomograms and conducted subtomogram averaging. Using cryo-ET and subtomogram averaging, they provided beautiful geometrical description of the central hub of cartwheel, which is made of the SAS-6 protein, one key of nine-fold symmetry, as well as the joint between the cartwheel and the triplet microtubule from *Paramecium* and *Chlamydomonas* and discussed structural differences between these two species. Regarding *Naegleria* and human, they compare length of the cartwheel and areas of the triplet decorated by different structural features.

This reviewer would recommend the manuscript for publication, once 3D structure of the cartwheel central hub and spokes from *Naegleria* and human are reconstructed by subtomogram averaging for further comparative structural studies and several technical issues mentioned below are clarified. This paper after addressing these points will attract the wide readers' attention.

****Major points****

1. Subtomogram averaging of human and *Naegleria* cartwheels and triplets

The authors stress evolutionary insights obtained in this work (Line 471). Subtomogram averaging of human and *Naegleria* centrioles, both for the central

hub and the triplet at the cartwheel region is necessary. Especially human cartwheel structure is awaited and thus will make this work worth for publication in a Journal of high reputation. Is the periodicity and conformation of the hub and the spoke same as one of the three unicellular organisms? Do they find any novel structural feature in addition to Greenan et al. reported? How about structure inside the hub?

We understand the reviewer's impatience and curiosity for the human cartwheel structure, which we share entirely. However, this remains extremely challenging due to technical limitations of the human sample. Indeed, as previously shown in Guichard et al 2010 (doi: 10.1038/emboj.2010.45.), Greenan et al 2019 (DOI: 10.7554/eLife.36851, discussed in the reviewer response) and Le Guennec et al 2020 (DOI: 10.1126/sciadv.aaz4137), purified human centrioles become extremely compressed during cryo-electron microscopy preparation (stated in the manuscript pages 8-9 (underlined)). High-resolution subtomogram averaging is not possible on such a deformed sample. Moreover, the human cartwheel is even more difficult to obtain than cartwheels from other species: mature human centrioles lack cartwheels (as stated in the manuscript page 4 (underlined)), and even when the cartwheel is observed inside rare procentrioles, it is very small compared to the *Trichonympha* cartwheel. We spent considerable effort acquiring a large dataset of 74 human centriole tomograms. However, only 8 tomograms contained procentrioles, 7 of which contained cartwheels with various amounts of compression. For these reasons, we do not provide such an analysis, even though we would love to do it.

Subtomogram averaging of the *Naegleria* centriolar proximal region will not be of high quality, as these centrioles are also compressed and broken (stated in the manuscript pages 8-9 (underlined)). Nevertheless, we performed subtomogram averaging of the proximal microtubule triplets of these centrioles, but with all the structural heterogeneity imparted by compression, it was not informative (see below).

Concerning the periodicity of the central hub in human and *Naegleria* centrioles, these measurements are described in the current version of the manuscript in Figure 2J and Figure S1. We indeed do not show the periodicity of the spokes in these species as we could not see them due to the compression of the centrioles. We prepared a new supplemental Figure S4 highlighting how the central hub periodicity was measured.

2. Periodicity of the cartwheel

The authors conducted subtomogram averaging by picking particles with 25nm distance (Line843, Line860). This will lead the following analysis to the 25nm periodicity assumed and may cause a risk of structure not following 25nm periodicity being smeared out. Therefore longitudinal periodicity must be carefully confirmed. Fourier analysis (such as Fig.S3CF of the other manuscript, Nazarov et al.) is one way to prove periodicity in unbiased way, but might be difficult with short cartwheels. This reviewer would propose classification of randomly extracted (but of course along the cylindrical hub or along the triplet microtubules) subtomograms. This experiment will end up with multiple subaverages, which are 25nm (or multiple times of that) shifted from each other. Then it will prove their assumption. The alternative way is to average assuming long enough periodicity, which is the least common multiple of all the possible periodicity. This reviewer is not optimistic, since the cartwheels studied in this work is relatively short.

We agree with the reviewer that, in principle, classification of randomly extracted subtomograms would be ideal. However, once again owing to the short length of the cartwheel and thus the very few boxes that can be extracted, this method cannot be applied here. Moreover, as pointed out by the reviewer, extracting longer boxes is also not applicable here, as the cartwheel is too small.

However, we performed a translational analysis on the *P. tetraurelia* and *C. reinhardtii* cartwheel spokes to confirm the observed periodicities, a strategy that we previously applied successfully on the *Trichonympha* cartwheel (Guichard et al 2012, DOI: 10.1126/science.1222789). This new data is now presented in Figure S14 and explained in detail in the material and method section (page 45 of the revised manuscript (highlighted in red)).

We also tried to perform a Fourier analysis as we did in the Guichard et al 2012 paper. However, in contrast to *Trichonympha*, which owing to its exceptional length led to a beautiful description of the cartwheel, we are working with a limited number of cartwheel repeats in this case (roughly 3), which means that this method did not lead to a significant or convincing signal, in contrast to the successful translational analysis described above.

Finally, we are confident that the 25 nm periodicity that we report is correct, as it can be seen directly in the raw tomograms. This very important analysis is shown in Figure S8 and is referred to as in page 12 of the revised manuscript (underlined).

The authors also need to clarify if the periodicities of the central hub from *Chlamydomonas* in this work (4.0nm in Line135), their original paper (8.5nm), CID (8.7nm in Line142) and the microtubule triplet correlate each other or are independent. If they are independent, heterogeneity will come from this periodicity mismatch. Image classification of parts with masking other part of the volume will clarify this.

We understand the reviewer's concern and are grateful for this comment. This apparent periodicity mismatch is mainly a technical limitation due to the limited resolution of the bin2 tomograms (the distance spanned by two bin2 pixels is ~1.4 nm). This means that we observe a technical variability in our measurements that brings some heterogeneity. Such a phenomenon has been already reported, for example in the case of the cilia, where periodicities of 8.21-8.35 nm and 8.3 +/- 0.5 nm have been detected at random intervals along the axoneme (Ma et al. <https://doi.org/10.1016/j.cell.2019.09.030> and Ichikawa et al. <https://doi.org/10.1073/pnas.1911119116>, respectively).

3. Classification

They employed multi-reference image classification (Line820-822) to detect polymorphism inside the cartwheel, assuming that there are two conformations coexisting and there are two conformations depending on the proximity. What is the basis of this classification strategy? What kind of volume was used as a template? The distribution of the multiple structures should be shown (like Fig.S5F from Nazarov et al. 2020, another paper of them).

We apologize for not having been sufficiently clear regarding this point. First of all, this method has been applied on the A-C linker and not the cartwheel. Second, we chose this strategy based on the changes of the connection to the microtubule triplets that are reported in Figure 4. Indeed, we observed that the A-C linker, pinhead and triplet base structure were not always present along the proximal region of the microtubule triplet. This phenomenon was also observed in the Li et al 2019 paper (doi: 10.7554/eLife.43434), where they demonstrated that the A-C linker is a flexible structure that gives the centriole a diaphragm-like

motion along the proximal region. We therefore decided to discriminate between the very proximal region to the more distal proximal region.

Finally, concerning the template used: we actually did not use a structured starting template but instead based our analysis on the gold standard reference-free method. In brief, we performed a multi-reference alignment using a random set of particles as a starting point to ensure that we were not introducing bias. We next divided the data into two independent half sets for final alignments and resolution estimation. A better description of this analysis can now be found in the revised methods section (page 47-48 of the manuscript, in red).

Lastly, as suggested, we included a supplemental figure (Figure S12) showing the distribution of the subtomograms after classification, similar to the co-submitted paper.

4. Quality of Naegleria and human tomograms

Quality of tomograms from *N. gruberi* (Fig.2B, in which the central hub is broken) and human (Fig.2C) is not good enough to discuss cartwheel geometry in a convincing way. This reviewer imagines that cartwheels were damaged during preparation from *N. gruberi* (as happened in Li et al. 2012 and improved by them Guichard et al. 2013). Preparation with shorter time will help. Fig. 2B (and other images with similar quality) must be replaced by better images and data from this tomogram should be excluded from the statistics in Fig.2J and others.

We thank the reviewer for raising this important point. It is indeed essential to understand how we could extract geometrical features from the broken *Naegleria* and human centrioles. We now provide a new supplemental Figure S4 to explain this in detail, showing how we only analyze intact portions in both cases.

Finally, we decided not to replace the *Naegleria* Fig. 2B with a non-broken cartwheel, as we wanted to show a representative image to illustrate the fact that many *Naegleria* centrioles were broken. We now mention this in the legend of Figure 2B (page 26 of the revised manuscript in red).

5. Quality of subtomogram averaging

Image quality of the averaged subtomograms is not so high as expected from FSC-based resolution estimation (similar to the resolution of Li et al. 2019), judging from appearance (separation of adjacent tubulins) of the doublet microtubules. What is the reason? Can their way to calculate resolution with

0.143 cut-off be considered as gold standard FSC? For gold standard FSC, once separated subtomograms must be aligned, averaged and further iterated without cross-talk. After extraction by Dynamo, which program did they use for subtomogram analysis (the other paper used Relion, thus gold standard FSC was properly calculated, this reviewer believes)?

Once again we have to apologize if we were not clear enough concerning the FSC-based resolution estimation.

In Li et al 2019, the estimated resolution of the microtubule triplet maps were around 22Å using a 0.143 cut-off, while our maps are around 31-37 Å (Figure S11). This difference might explain the discrepancy in image quality as mentioned by the reviewer.

We agree with the reviewer that the gold standard FSC should be performed using independent subtomogram half sets, and this is exactly what we did our analysis. This is reported on page 47-48 of the manuscript (in red).

Concerning the subtomogram alignment and averaging, we indeed used Dynamo, which similar to Relion, separates the dataset in two half sets that are aligned in parallel independently. From these two independent averages, we then computed the “gold-standard” FSC using the EMAN2 package (Tange et al., 2007). In the revised manuscript, we clearly outline this workflow and the software used (page 47-48 of the manuscript (in red)).

6. Cartwheel protrusion

Does the proximal cartwheel protrusion (Fig. 2KL) have similar spoke structure as the main part of the cartwheel? They modeled as if the cartwheel spoke has the same conformation there as in the main part of the cartwheel (Fig.6), but without experimental basis. By averaging subtomograms from the cartwheel, the authors will see if the conformation of the spoke is determined by the triplet microtubule binding or stable by itself.

We agree with the reviewer that this is an interesting issue that we had not explored enough. As presented in Figure 2K, the cartwheel protrusion in *P. tetraurelia* centrioles represents only ~15% of the total cartwheel length. The number of sub-volumes is then too small to perform reliable averaging. We therefore applied symmetrization to volumes with cartwheels sticking out, in both *Paramecium* and *Chlamydomonas*. As presented in the new Figure S6, we found that the cartwheel protrusion has similar spoke structure/organization to the MTT-attached cartwheel. Moreover, we identified that the ends of the spokes are connected together via a rod-like linker that, based on our measurements, corresponds to the D2 densities. We therefore added this new “D2-rod” linker to

Figure S6 and the summary model presented in Figure 6 and added this information page 10 of the revised manuscript (in red).

We thank the reviewer for this suggestion, which brings a great addition to our manuscript.

****Minor points:****

Line 233: green dotted lines do not exist in Fig.3DJ. There should be more visible way to present "slightly tilted spokes", for example by a movie.

We added a white dotted line on the revised versions of Figures 3D and S7A.

Concerning the visualization of tilted spokes, we have added a Figure S13 with the full volume of the *P. tetraurelia* cartwheel as well as a supplementary movie S1.

Line 237: Periodic features are visible in the right panels of Fig.S6AB, but they seem not to be raw data, but after some operation (not written in the legend; please clarify). The central panels are the raw data, but periodic feature is not clear in the top center panel.

We apologize for this mistake. What is presented in Fig.S6AB (now Figure S8 in the revised manuscript) corresponds indeed to raw data that were filtered using the "Gaussian filter 3D" filter from Fiji. We then projected slices of ~50 nm, 25 nm and 8 nm in thickness, in order to increase the contrast. We clarified this in the legend as requested (page 38 of the revised manuscript in red).

Concerning the periodicities, it is indeed difficult to see the periodicities of the spokes and the hub in the same section through the tomographic volume. This is why we have shown different images where we can see the spokes (middle panel) and the hub periodicities (right panel). We added a sentence in the legend (page 38 of the revised manuscript in red).

Line 440: triplet base

They attempted to interpret molecular identification of the triplet base. It is fine to discuss/model, but too detailed discussion only based on molecular weight and coiled-coil prediction will not add any reliability to this model, since the composition of this area is not known. To locate proteins, either deletion mutant analysis, label (such as antibody) or genetic tagging, or biochemical methods (such as cross-linking MS) is necessary. This reviewer recommends removal of too detailed statement (Line 442-449).

We agree with the reviewer that this is just a discussion point and that it is not proven yet. We feel that such a hypothesis is interesting and belongs in the discussion and stated clearly in the revised text that this is an hypothesis (p.21 in red).

Line 555 (Fig.3AB): it is confusing to use yellow dotted line for two different objects in Fig.3A and B. it would be more convenient to use different colors.

We fix this in the revised version of the manuscript using light orange and yellow.

Fig.3I: a similar surface rendering presentation should be shown for *Chlamydomonas* data as well.

We agree with the reviewer and have included this rendering representation in Figure S7 (panel C) of the revised version of the manuscript.

Reviewer #1 (Significance (Required)):

This reviewer would recommend the manuscript for publication, once 3D structure of the cartwheel central hub and spokes from *Naegleria* and human are reconstructed by subtomogram averaging for further comparative structural studies and several technical issues mentioned below are clarified. This paper after addressing these points will attract the wide readers' attention.

We understand that the reviewer is very eager to see the human and *Naegleria* averaging. However, as explained above, this is unfortunately unreachable with the current dataset. Regardless, we believe that our manuscript's analysis of cartwheels from four new species, including the first in situ look at the cartwheel structure inside native cells, represents a big step forward in understanding the organization and conservation of the centriole proximal region, paving the way for future studies.

Reviewer #2 (Evidence, reproducibility and clarity (Required)):

In this manuscript Klena et al., use cryo-electron tomography (CET) to analyse the structure of the proximal end of centrioles and procentrioles. These tiny organelles have many important functions in cells so there is considerable interest in understanding how they are assembled and function. While there has been great progress in defining the proteins required for centriole assembly/function, their small size has meant we have very limited information about their detailed structure. The Hachet/Guichard team have been applying CET methods to this problem for some time (initially with Pierre Gonczy), and in this manuscript they use the most up-to-date technology to analyse centriole structure in several species-comparing this to the lower resolution structures from *Trichonympha* they published previously (Guichard et al., *Science*, 2012; Guichard et al., *Curr. Biol.*, 2013).

I am not an expert in CET, so it will be important to have a review from someone with this technical knowledge (although this team are real experts, so I would not anticipate any problems). As a biologist, the data looks very impressive; really beautiful images and a compelling narrative that was generally well presented and easy to follow. I found this paper fascinating, and these structures provide several important insights that will profoundly influence our thinking on how these organelles assemble and function. Several aspects of the overall architecture are now shown to be widely conserved-such as the extension of the cartwheel below the centriole MTs, the CID, and the architecture of the linkage between the cartwheel and the MTs (although several groups have recently published quite similar data focused on the architecture of the centriole MTs). Equally important are those aspects that seem to vary between species, most surprisingly the organisation of the cartwheel spokes that link the inner hub to the outer MTs. These findings will be of great interest to the centriole and centrosome fields and I strongly support publication in *The EMBO J*. I have only a few points that the authors should consider.

We thank the reviewer for his/her enthusiasm towards our manuscript.

1. Perhaps the most important is the relationship of this paper to the Nazarov et al., paper that was co-submitted by the Gonczy laboratory. The two papers complement each other well, and I strongly support back-to-back publication. However, the papers come to slightly different conclusions on a number of points, and it would be very useful to the field if these could be addressed/discussed so we can hear the author's thoughts on these points.

We thank the reviewer for this comment and we would be happy to discuss the points raised below.

1A. The first issue is cartwheel polarity. The authors do not make much of polarity here, but they conclude that the spokes emanating from the hub are asymmetric (Fig. 3I). They say this is true for both *P. tetraurelia* and *C. reinhardtii*, but only show this for the former and it looks much less clear for the latter-so this should be shown properly, or at least discussed in more depth. On the basis of this they conclude that the core 25nm repetitive unit of the cartwheel has intrinsic polarity. Nazarov *et al.*, make the same conclusion, but base this on the asymmetric localisation of the CID within the hub, something that is not reported here. This may be due to species differences, but it is important to know whether these authors have looked for this and what they conclude.

Following the reviewer's advice, we now include the *C. reinhardtii* surface rendering of the cartwheel map to show that while we detect the spoke asymmetry, it is less pronounced than that of *P. tetraurelia* cartwheel (Figure S7C).

We did not observe the asymmetric localization of the CID reported by Nazarov *et al.* In our manuscript, we observe heterogeneity in the CIDs of *Paramecium* and *Chlamydomonas* (as shown in Figure 3C and 3J). We therefore do not feel confident in making a conclusion about this. We added a sentence to page 19 in the revised manuscript related to this comment (in red).

1B. The second issue is that Nazarov *et al.*, provide evidence that strongly suggests that in some species the double stacked Sas-6 rings are offset from one another. Again, it would be important to know if Klena *et al.* have looked for this? Perhaps the staggering of the 3 spokes in the *P. tetraurelia* structure could be interpreted in this way (Figure 3J)?

As suggested by the reviewer, we checked whether we can detect the reported offset, and indeed could see it in our subtomogram average of the *Paramecium* cartwheel. Interesting, the offset is opposite to the one described in the accompanying Nazarov.*et al.* manuscript. We undertook to make the fit of a pair of SAS-6 rings in our best map, the *P. tetraurelia* one (the resolution of the *Chlamydomonas* map was not sufficient for this kind of approach). Unfortunately, the *Paramecium* central hub is larger than the LmSAS-6 ring structure. It would be necessary to stretch the proteins to fit the two structures, which we decided not to do because it could create a false conclusion. We now provide a new Figure S13 to illustrate this point, and we discuss this point on pages 19 of the revised manuscript (in red).

1C. Finally, I was surprised that the authors did not try to model the crystal structure of Sas-6 into their various cartwheel models, especially in *C. reinhardtii* where the Sas-6 structure is known (but also for the other species using the *C. reinhardtii* structure, which is highly conserved). It perhaps seems obvious how the Sas-6 rings might be arranged in these structures, but this overall organisation of the spokes is quite surprising (and quite different to how originally envisaged in the more regular *Triconympha* structure). Nazarov et al., perform this sort of analysis, and the Sas-6 structure fits as one might have expected, but this analysis provides some support for the offsetting of the stacked rings and I would be very interested to see it here.

The point raised by the reviewer is along the same lines as the reported offset. We now provide a new Figure S13 with fitting SAS-6 rings similarly to *Nazarov et al.* However, as explained above, the hub diameter of the *P. tetraurelia* cartwheel is different from those of the only known SAS-6 (LmSAS-6) ring structure (others are models), so we cannot propose a precise fit without artificially stretching the molecules. Furthermore, as mentioned in the discussion, we cannot exclude that another protein might be involved in this organization. This is now included and discussed in the revised manuscript on pages 19-20 (in red).

2. To the non-expert eye it is quite difficult to understand how meaningful data can be extracted from images that look like the human cartwheel presented in Fig.2C. I don't doubt that this is the case, but it might be worth briefly describing how this was done in the legend (or pointing to a section in the Materials and Methods if a slightly longer explanation is required). One can see that the human data is more variable, but it would be good to know how the authors decided what was suitable for analysis (perhaps, for example, only the CID highlighted with an arrow was counted in this image?).

We completely agree with the reviewer, and we think this is an essential point. We now include a supplemental Figure S4 to explain how we could extract meaningful information on additional examples from *Naegleria* and Human broken centrioles. As explained to reviewer 1 (point #4, above), we only analyze intact portions of these centrioles. This limited our analysis to intact regions of the cartwheel hub, as the spokes were disrupted by the centriole compression.

Reviewer #2 (Significance (Required)):

See above.

Reviewer #3 (Evidence, reproducibility and clarity (Required)):

****Summary****

In this manuscript, Klena and colleagues describe the high-resolution architecture of centrioles from four different species using cryo-electron tomography. They mainly focus on the cartwheel-containing region which is important for early events of centriole duplication. The authors revealed several conserved features in four different species (*Chlamydomonas reinhardtii*, *Paramecium tetraurelia*, *Naegleria gruberi*, and human). For example, the authors show that the periodicity of the cartwheel central hub is about ~4 nm, which was previously shown to be 8.5 nm from *Trichonympha*. They also determined the cartwheel inner density (CID) and proximal protrusion of cartwheel in all investigated species. The authors should consider the following:

****Major point****

Line 263 to 276: To solve the discrepancy in central hub periodicity between *Trichonympha* and *P. tetraurelia*, the authors lowered the resolution of images from *P. tetraurelia*. However, it would be better to show high resolution images of *Trichonympha* to support this possibility.

We thank the reviewer for his/her positive comments. To answer this point, on page 13 (in red), we now cite the work of Nazarov et al., which reports the same periodicity in their high-resolution structure of the *Trichonympha* cartwheel central hub that we described in the *Paramecium* central hub.

Reviewer #3 (Significance (Required)):

Their experimental results are of very high quality and provide clues as to which organizational features of their architectures are conserved or not. I believe this manuscript would be interesting for anyone who is researching about the centrosomes. Therefore, I would like to recommend this paper to be published in the EMBO journal which is affiliated with the Review Commons.

Once again, we thank the reviewer for his/her positive comments.

Prof. Paul Guichard
Université de Genève
Cell Biology Department
Faculté des Sciences, Sciences III
30, quai Ernest-Ansermet
Geneve 1211
Switzerland

31st Jul 2020

Re: EMBOJ-2020-106246

The architecture of the centriole cartwheel-containing region revealed by cryo-electron tomography

Thank you again for submitting your revised Review Commons manuscript to for EMBO Journal consideration. In light of the positive previous comments and the interest of the subject of the study, I decided to treat it essentially as a revision, sending it back to referee 1 for assessing your responses to the various technical concerns originally raised by this reviewers. As you will see from the comments copied below, although the referee would have desired more detailed new information from sub-tomogram imaging especially for human centrioles, they appreciate the technical difficulties and remain nevertheless supportive of publication. We shall therefore be happy to accept this work for The EMBO Journal, pending incorporation of various editorial points as detailed below.

I am therefore returning the manuscript to you for a final round of minor revision, to allow you to make these adjustments and upload all modified files. Once we will have received them, we should be ready to proceed with formal acceptance and production of the manuscript.

Hartmut Vodermaier, PhD
Senior Editor / The EMBO Journal
h.vodermaier@embojournal.org

REFEREE 1:

The authors addressed points raised by the reviewers well. Instead of the way this reviewer proposed, they applied translational analysis to prove 25nm periodicity of the centriolar spokes. Since they did not provide subtomogram averaging of centrioles from human and Naegleria due to technical difficulty (they stated this in the text), the essential new knowledge of this work is about unicellular organisms. Nevertheless this work will trigger further studies at molecular level. This reviewer supports publication of this manuscript.

DÉPARTEMENT DE BIOLOGIE CELLULAIRE

Paul Guichard

Prof. Paul Guichard
University of Geneva
Switzerland

Ref: EMBOJ-2020-106246

August 20th 2020,

Following your message dated August 11th, we are happy to see that you found our revised manuscript improved and that you deemed our manuscript ready for publication in EMBO Journal.

As requested, we have modified the manuscript to fit the EMBO J guidelines and created 5 EV figures as well as Appendix supplementary Figure. We have also made the requested changes in the main manuscript keeping the “Track changes” option to ease the subsequent editing and reduced the number of keywords.

You will find as well together with the revised manuscript a synopsis with bullet points as well as a simplified schematic image. You also would like to propose two possibilities of images for a Cover, please let us know in case you are be interested.

With these changes, we hope that you will find our manuscript fully ready for publication in EMBO Journal. Do not hesitate if you have further questions/remarks.

Sincerely,

Virginie Hamel

Paul Guichard

Benjamin Engel

Thank you for submitting your final revised manuscript for our consideration. I am pleased to inform you that we have now accepted it for publication in The EMBO Journal.

Corresponding Author Name: Paul Guichard, Virgine Hamel, Benjamin Engel

Manuscript Number: EMBOJ-2020-106246